# Exploration of the dynamic interplay between lipids and membrane proteins by hydrostatic pressure

Alexandre Pozza[1,6], François Giraud[2,6], Quentin Cece[1,4], Marina Casiraghi[1,5], Elodie Point[1], Marjorie Damian[3], Christel Le Bon[1], Karine Moncoq[1], Jean-Louis Banères [3], Ewen Lescop [2✉] & Laurent J. Catoire [1✉]

Cell membranes represent a complex and variable medium in time and space of lipids and proteins. Their physico-chemical properties are determined by lipid components which can in turn influence the biological function of membranes. Here, we used hydrostatic pressure to study the close dynamic relationships between lipids and membrane proteins. Experiments on the $\beta$–barrel OmpX and the $\alpha$–helical BLT2 G Protein-Coupled Receptor in nanodiscs of different lipid compositions reveal conformational landscapes intimately linked to pressure and lipids. Pressure can modify the conformational landscape of the membrane protein per se, but also increases the gelation of lipids, both being monitored simultaneously at high atomic resolution by NMR. Our study also clearly shows that a membrane protein can modulate, at least locally, the fluidity of the bilayer. The strategy proposed herein opens new perspectives to scrutinize the dynamic interplay between membrane proteins and their surrounding lipids.

[1] Laboratoire de Biologie Physico-Chimique des Protéines Membranaires, UMR 7099, CNRS/Université de Paris, Institut de Biologie Physico-Chimique (IBPC, FRC 550), 75005 Paris, France. [2] Institut de Chimie des Substances Naturelles (ICSN), CNRS UPR 2301, Université Paris-Saclay, 91198 Gif-sur-Yvette, France. [3] Institut des Biomolécules Max Mousseron (IBMM), Université de Montpellier, CNRS, ENSCM, Pôle Chimie Balard Recherche, 34293 Montpellier, cedex 5, France. [4] Present address: Laboratoire Cibles Thérapeutiques et Conception de Médicaments (CiTCoM), UMR 8038, CNRS/Université de Paris, Faculté de Pharmacie, 75270 Paris, Cedex 06, France. [5] Present address: Department of Molecular and Cellular Physiology, Stanford University School of Medicine, 94305 Stanford, CA, USA. [6] These authors contributed equally: Alexandre Pozza, François Giraud. ✉email: ewen.lescop@cnrs.fr; laurent.catoire@ibpc.fr

Cells, and organelles such as the nucleus, the endoplasmic reticulum, or the Golgi apparatus, are delimited by membranes that display a great variety in lipids and proteins to perform numerous fundamental functions. Apart from a few exceptions, lipid membranes are crowded in proteins[1], and both lipids and proteins, i.e., not only lipids, shape the geometry and the physicochemical properties of membranes which in turn regulate protein function[2,3]. Among these properties, the membrane fluidity resulting from the mobility of lipids and the degree of concentration of proteins represents a fundamental parameter intimately associated with vital functions to maintain the homeostasis of cells across organisms whose alteration can result in many metabolic disorders[4–6]. In poikilothermic organisms, whose internal temperature can vary to adapt to environmental stress, the lipid composition is continuously adapted upon variations in temperature to preserve the fluidity of membranes[7] or in the control of the respiration metabolism[8]. In the piezosphere, which comprises deep marine environments and the lithosphere, piezophilic organisms can withstand high pressures by adapting in particular their lipid composition known as homeoviscous adaptation[7] where an increase in the unsaturated to saturated lipid ratio is thought to preserve functional membrane fluidity and permeability in piezophiles at high pressure[9]. Lipids are also closely involved in the opening and closing of mechanosensitive channels in bacteria, fungi, plants, and animals[10]. At physiological temperatures and pressures, biological membranes are supposed to be fluid, but this does not exclude local important variations in fluidity. The viscosity can indeed be impacted by protein crowding[1] or considering the existence of sufficient amounts of saturated lipids or sterols[11] (e.g., the cholesterol/total phospholipids molar ratio is ~1 in mammal plasma membranes[12]).

Here, we investigated at the atomic scale the impact of the gelation of lipids on the conformational ensemble of the prokaryotic $\beta$-barrel Outer membrane protein X (OmpX) from *Escherichia coli* (*E. coli*)[13] and the human $\alpha$-helical leukotriene BLT2 G Protein-Coupled Receptor (GPCR)[14] and conversely, the impact of these proteins on lipid dynamics. To do so, we advantageously associated lipid nanodiscs[15,16] with high hydrostatic pressure solution-state NMR spectroscopy. Pressure is a fundamental thermodynamic variable for studying both conformational equilibria and protein dynamics[17–20]. Biological membranes are exquisitely sensitive to phospholipid composition and pressure variations. This pressure variation results in a complex Pressure/Temperature (P/T) phase diagram and the main gel-fluid phase transition temperature is shifted down by −22 °C/kbar for saturated phosphatidylcholines[21,22]. Pressure then offers the possibility to control lipid bilayer dynamics.

NMR experiments have been performed in a pressure range (1–2500 bar) where both membrane proteins (MPs) studied herein have most of their structures unperturbed (vide infra). From a physiological point of view, this pressure range is in the same order of magnitude as those experienced by obligate piezophiles and piezotolerant organisms, e.g., at 1250 bar for an archaeon isolated from a deep-sea hydrothermal vent[23]. Our strategy resides in comparing pressure effects on different nanodisc compositions to discriminate lipid-dependent or direct pressure effects on the perturbation of protein conformational landscapes and to assess the impact of MPs on the dynamics of the surrounding lipids.

## Results and discussion

In the function of the MP under study, we used different lipid compositions organized as small nanometric bilayers, i.e., nanodiscs (cf. Methods section). The low molecular fast tumbling nanodiscs ensure highly resolved NMR signals and the high protein:lipid ratio in the particle prevents the dilution of the effect of the MP into too many lipids. With the prokaryotic outer MP OmpX, we chose 1,2-dimyristoyl-sn-glycero-3-phosphocholine DMPC (transition temperature $T_m = +24$ °C at 1 bar) considering that the outer membrane of *E. coli* is predominantly populated with lipids with acyl chains made of 14 carbon atoms. In addition, OmpX embedded in DMPC nanodiscs has been extensively studied by NMR[24]. With the eukaryotic receptor, we used two different lipid compositions with different lipid dynamics: (i) a mix composed of phosphatidylcholine POPC associated with phosphatidylglycerol POPG ($T_m = −2$ °C at 1 bar) and, (ii) dipalmitoylphosphatidylcholine DPPC ($T_m = +41$ °C at 1 bar), both including 1 mol% of cholesterol to stabilize the receptor[25]. In the following, these nanodiscs will be referred as to POPC/POPG and DPPC nanodiscs without explicitly mentioning cholesterol. Although DPPC is expected to be in the gel phase at 25 °C over the 1–2500 bar used here, POPC/POPG mixture is expected to undergo fluid-to-gel phase transition ~1400 bar. In the case of DMPC nanodiscs, the NMR experiments were performed at 25 °C and 40 °C up to 1750 bar in order to collect data at a different degree of fluidity of the bilayer at a given pressure (transition pressures $P_m = ~$few tens of bar and ~800 bar at 25 °C and 40 °C, respectively[26]). Note that the lipid P/T diagram was derived for large lipid structures and to our knowledge how nanodiscs and the embedded MPs react to pressure is yet unknown.

**Barotropic behavior of lipid nanodiscs**. In addition to provide a membrane-like environment and being adjustable in lipid composition, the nanodisc supramolecular structure resists high pressures in the absence of embedded MPs, as indicated by dynamic light scattering (DLS), size exclusion chromatography (SEC), and NMR experiments before and after a pressure jump to 2500 bar (Fig. 1). By applying a pressure ramp, we observed a progressive evolution of the NMR signals of both lipoprotein and lipids (Fig. S1) and we observed that lipoprotein and lipid NMR signals were collected at atmospheric pressure before and after the ramp were identical (Fig. 1). This opens the way to perform thermodynamic analysis of conformational transitions. The same observation is made in the presence of MPs (vide infra). For the three lipid compositions, the intensity of lipoprotein NMR signals decreases slightly along the pressure ramp, but to a much lesser extent compared to the lipid signals (Fig. S1). This evolution might reflect a slight reversible conformational change of the lipoprotein due to a decrease of the surface and volume of the lipid bilayer upon compression[26], to an acceleration of the intrinsic water/$^1H^N$ exchange rate under pressure[27] and/or to a slight increase of the pH of the solution even though Tris buffer is considered as quite bororesistant[28]. The reversible pressure deformations of nanodiscs thus open the possibility to use pressure as a variable to control the phase and dynamic behavior of lipid bilayers and the conformational landscape of embedded proteins.

Applying pressure is equivalent to cooling down the temperature of the lipid bilayer even though temperature and pressure do not the same effect on the phase behavior of lipids[29]. Upon pressurization, the acyl chains and heads of the phospholipids tend to be more ordered by adapting their shape to the diminution of the void volume between lipids[26]. At the transition pressure, these acyl chains switch from disordered conformations, that correspond to the fluid or liquid-crystalline (LC) phase, to more extended and ordered conformations which characterize the gel phase. Lipid dynamics in the gel and LC phases are sufficiently different to impose a dramatic change in $^1H$ lineshape/intensity

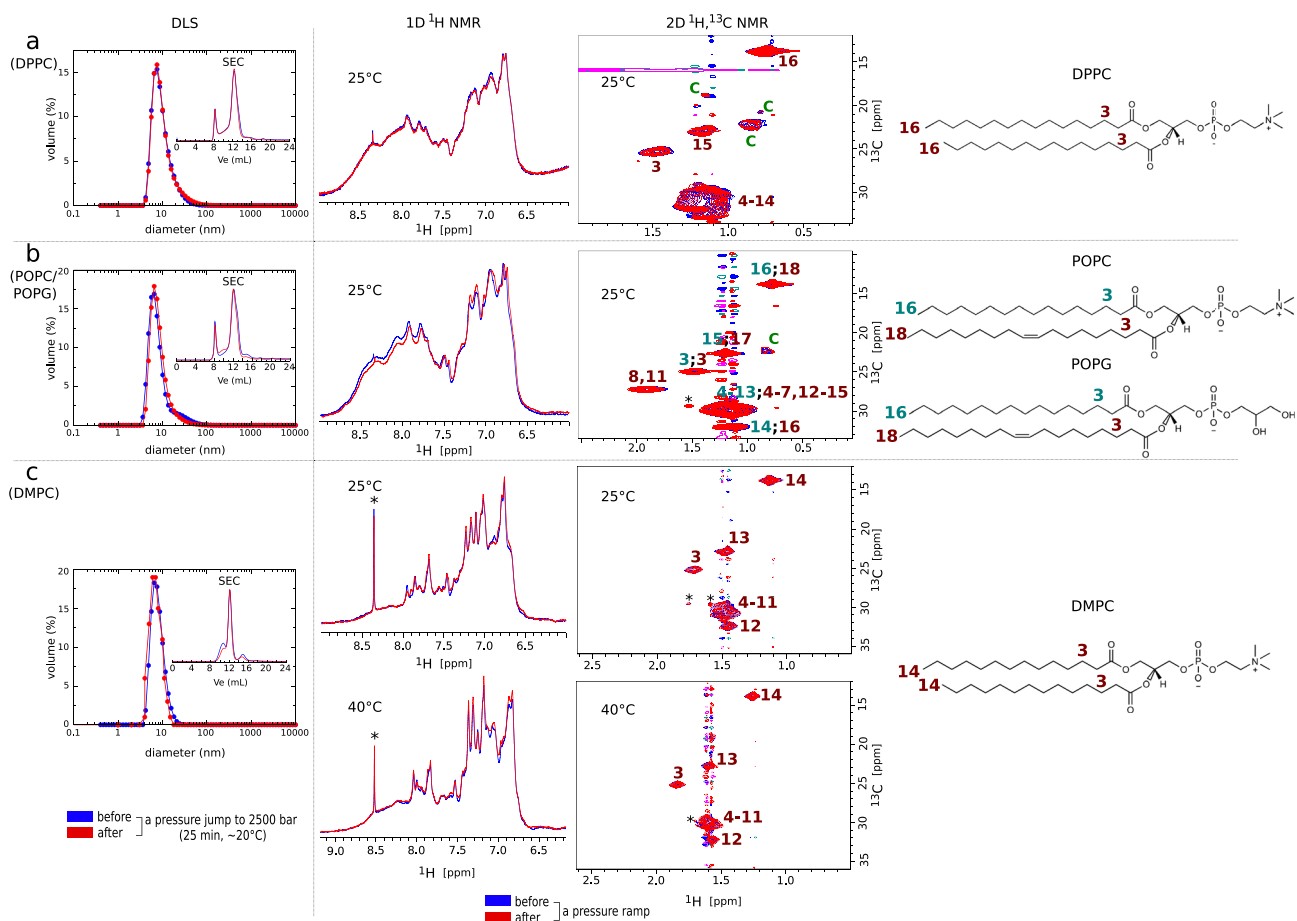

**Fig. 1 Barotropic behavior of lipid nanodiscs.** Illustration of the reproducibility of DLS, SEC, and NMR data (**a**: DPPC; **b**: POPC/POPG; **c**: DMPC) of empty, i.e., without MP, nanodiscs collected at atmospheric pressure before and after a pressure jump to 2500 bar during 25 min (DLS and SEC experiments) or ramp (NMR experiments) (see Methods). For DLS and SEC experiments, the graphs represent the volume-weighted distribution function of the diameter of nanodiscs in nm and the absorbance at 280 nm in the function of the volume of elution (the y axis has been removed for clarity), respectively. One-dimensional (1D) $^1$H NMR spectra represent a close-up in the region of amide, amine, and aromatic protons of the lipoprotein MSP1D1 and 2D $^1$H,$^{13}$C spectra are plotted in the CH$_2$ and CH$_3$ resonance region of the lipid acyl chains. The numbers on the 2D NMR spectra indicate the CH$_n$ (n = 2 or 3) moieties composing the lipid acyl chains indicated on the chemical structures on the right (NMR assignments are based on abundant literature, e.g., ref. [66]) and the letter C in green indicates some signals of the cholesterol in the case of POPC/POPG and DPPC nanodiscs. The asterisk symbols in 1D and 2D experiments indicate residual traces of imidazole and weak signals that could correspond to CH$_2$ moieties spatially close to the lipoprotein, respectively.

of phospholipid NMR signals thus allowing direct detection of the gel/LC transition[17].

In the absence of MP, by increasing the pressure we observed a fluid-to-gel phase transition for lipid acyl chains ~800 bar at 40 °C for DMPC (Fig. 2a) and 1400 bar for POPC/POPG at 25 °C (Fig. 2e). Hence, the phospholipids confined in the nanodiscs display barotropic phase transition P/T values similar to observations based on larger-scale systems made of pure DMPC or POPC[26,30]. At 1 bar, the transition temperature of large-scale DMPC bilayers is 24 °C[26]. Accordingly, when DMPC nanodiscs were subjected to pressure at 25 °C, we observed a continuous decrease in intensity without any clear transition in agreement with the continuous stiffening of the gel phase. The dynamics of DMPC and both POPC and POPG heads are less affected by pressure than the acyl chains except protons *g1, g2*, and *g3* at the glycerol connection and *α* protons which display substantially reduced dynamics upon the pressure ramp (Fig. 2a, e). Worthy of note, in the case of POPC/POPG nanodiscs, a pre-transition seems to occur at the head of the phospholipids just before the transition observed at the acyl chains (Fig. 2e), which could suggest that the heads stiffen before the tails upon pressurization.

DPPC acyl chains are in a gel phase at 25 °C at any pressure[26]. Nevertheless, the decrease in NMR signal intensity observed for lipid head and tails along the pressure ramp indicates that the mobility of DPPC is gradually reduced in the gel phase (Fig. 2g). The effect of pressure on the NMR signals is gradually increased from the lipophilic to the hydrophilic ends in the case of DPPC but not for POPC and POPG or DMPC. This is notably visible by comparing the absolute volumes of the NMR signals of the methylene groups in position 3 (V$_{CH_2(3)}$) vs. position 15 (V$_{CH_2(15)}$) along the acyl chains of DPPC measured from 2D $^1$H,$^{13}$C SOFAST-HMQC experiments (Fig. 3b). At atmospheric pressure, V$_{CH_2(3)}$ represents already only 70% of V$_{CH_2(15)}$ (not shown) and drops down to only 5% at 2000 bar and the signal is not observable anymore above this pressure. This indicates that a DPPC acyl chain is more mobile at the center of the bilayer than at the glycerol connection while POPC, POPG, and DMPC molecules display a more homogeneous dynamic behavior along their chains upon compression. To be noticed, pressurized DPPC nanodiscs are also capable to experience a fluid-to-gel phase transition at a $P_m$ of 500 bar when incubated at 50 °C for CH$_2$ groups (Fig. S2), i.e., slightly above the $P_m$ observed with

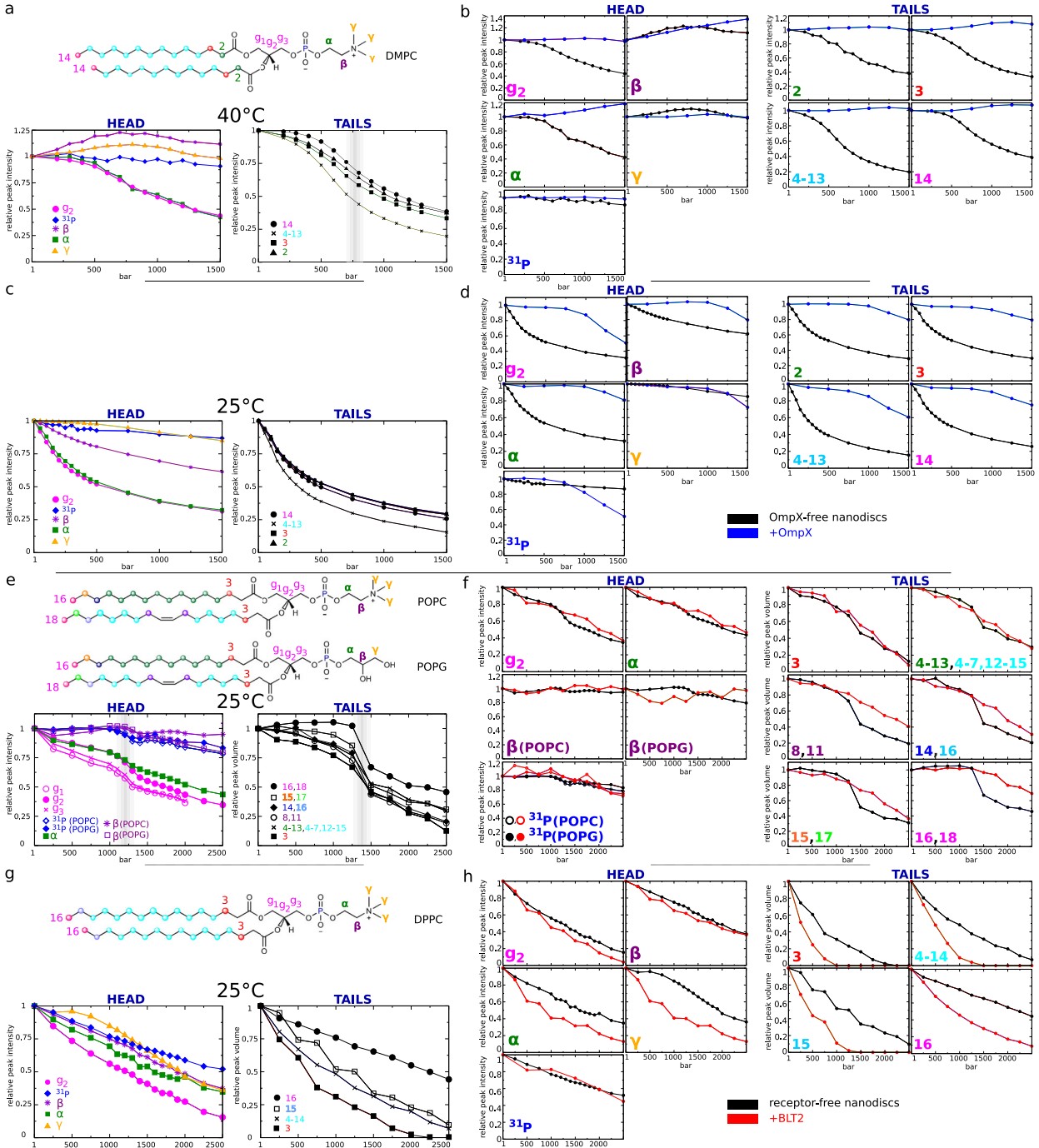

**Fig. 2 Barotropic behavior of lipid NMR signals. a**, **c**, **e**, and **g** describe the evolution of MP-free nanodisc NMR signals for, respectively, DMPC at 40 °C and 25 °C, POPC/POPG, and DPPC nanodiscs at 25 °C. In **b**, **d**, **f**, and **h**, the evolution of MP-free nanodisc lipid NMR signals (in black) are compared with OmpX-containing (in blue) or BLT2-containing nanodiscs (in red). Head and tail protons are shown separately. The color code indicated on lipid primary structures are reproduced in the labels of the graphs. The volume or intensity of reference for each signal has been taken at atmospheric pressure. The vertical gray bands indicate pressure-induced phase transition. γ protons of POPC and POPG are not represented because their NMR chemical shifts could not be unambiguously assigned. In the presence of MPs, it was not possible to measure the evolution of [1]H NMR signals of g1 and g3 for all lipids (see also spectra in Fig. S4 and S5). Source data are provided as a Source Data file.

liposomes for this lipid[26]. This confirms our ability by NMR to detect P/T phase transitions for nanodiscs and the existence of cooperativity between lipids for DPPC, DMPC, and POPC/POPG.

As observed with empty nanodiscs, DMPC and lipoprotein NMR signals were perfectly reproducible in the presence of OmpX at atmospheric pressure after a pressure ramp (Fig. 3a, b). Compared with empty nanodiscs, the presence of the β-barrel amplified some signals that probably correspond to lipids that interact with the belt of lipoproteins and/or OmpX (see asterisks in Figs. 1 and 3b). The presence of OmpX in the DMPC nanodisc considerably modifies lipid dynamics both at 25 °C and 40 °C. At

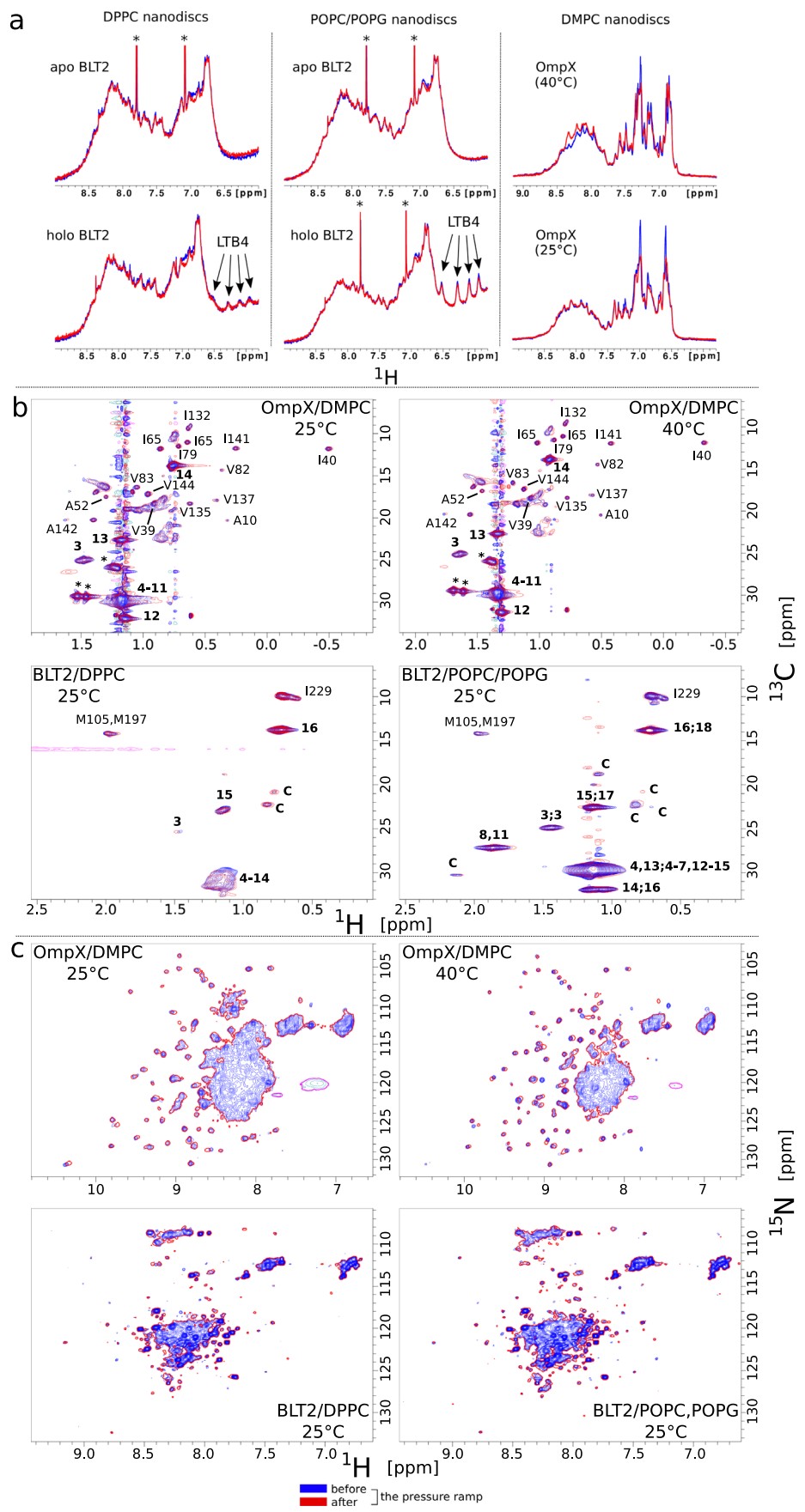

**Fig. 3 Stability of nanodiscs and their embedded MPs upon pressurization and depressurization. a** Illustration of the reversibility of amide, amine, and aromatic (lipoprotein and embedded MPs) NMR signals with DPPC (left), POPC/POPG (center), and DMPC (right) nanodiscs through 1D $^1$H spectra collected at atmospheric pressure. **b** Two superimposed 2D $^1$H,$^{13}$C-SOFAST-HMQC[47] spectra were collected at atmospheric pressure before and after a pressure ramp showing NMR signals of $^{13}$CH$_2$ and $^{13}$CH$_3$ lipid moieties and $^{13}$CH$_3$-$\beta$-Ala, $^{13}$CH$_3$-$\delta_1$-Ile, and $^{13}$CH$_3$-$\gamma_2$(proS)-Val of OmpX (top) or $^{13}$CH$_3$-$\epsilon$-Met and $^{13}$CH$_3$-$\delta_1$-Ile of BLT2 (bottom). In the case of OmpX, only methyl groups analyzed in this study are indicated (see Fig. 5a). The numbers in bold indicate the CH$_n$ ($n$ = 2 or 3) moieties composing the lipid acyl chains (see chemical structures in Figs. 1 or 2) and the letter C indicates some signals of the cholesterol (1 mol% of total lipids). The asterisks probably correspond to lipid signals that emerge in the presence of OmpX. **c** Illustration of the reversibility of backbone amide proton and nitrogen NMR signals of OmpX (top) and BLT2 (bottom) viewed in two-dimensional $^1$H,$^{15}$N-SOFAST-HMQC spectra. Spectra recorded after the pressure ramp in red are drawn with only one contour line to ease the comparison. The same stability has been observed for BLT2 in the presence of the agonist LTB4. For OmpX, the cluster of signals at the center of the spectra corresponds to residues belonging to extracellular loops that display some intermediate chemical exchange (pH ~ 7.3) at the NMR chemical shift timescales (see also Fig. S6a).

40 °C both DMPC head and tail proton signals display almost a constant intensity over the pressure range, instead of a clear transition, as observed in the absence of OmpX (Fig. 2b, d). Interestingly, because the gelation of DMPC at 25 °C occurs much earlier along the pressure axis compared with 40 °C, we finally observe a start of the transition at that temperature after ~1000 bar. We then concluded that OmpX did not reduce the transition cooperativity but instead shifted the transition to higher pressure. Taken together, this indicates that OmpX increases lipid dynamics under P/T conditions where lipids tend to be rigid (gel phase).

Similarly to MP-free or OmpX/DMPC nanodiscs, lipoprotein and either POPC/POPG/cholesterol or DPPC NMR signals in the presence of the $\alpha$-helical GPCR are fully reproducible at atmospheric pressure after a pressure ramp indicating that the presence of the integral MP does not disturb the stability of the supramolecular assembly during pressurization and/or a depressurization (Fig. 3a, b). In the presence of the receptor, the NMR signals of the lipid acyl chains are greatly impacted, as illustrated by the evolution of the peak volumes along the pressure ramp in Fig. 2f, h. In POPC/POPG nanodiscs, the presence of the receptor annihilates the LC to gel phase transition and delays the gelation process (Fig. 2f). In DPPC nanodiscs, the presence of BLT2 leads to the opposite effect, i.e., it accentuates the gelling of the acyl chains, and this effect seems to be evenly distributed along the chains (Fig. 2h). The dynamics of DPPC head are also reduced (Fig. 2g), while for POPC and POPG, only the $\alpha$ and $g_2$ protons are impacted. This is possibly due to the pressure-induced modulation of lipid-receptor interactions or of lipid dynamics. Hence, starting in an LC phase, upon the pressure ramp, the receptor tends to prolong the dynamic character of the liquid phase by delaying the gelation process, as observed for OmpX in DMPC, and starting in a gel phase, the receptor accentuates the stiffening of the lipids. In the case of a fluid phase, the effect of the receptor on the lipid phase seems analog to the order effect observed upon the addition of cholesterol to a phospholipid bilayer: the cholesterol decreases the area per lipid which results in an increase of the main-phase transition temperature ($T_m$)[11]. In the gel phase, the addition of the receptor may accentuate some changes in the physical properties of the lipid bilayer that already lead to gelation of the phase upon pressurization. This may concern a more pronounced increase of the membrane thickness or decrease in the lipid area and volume. For instance, the membrane thickness and hydrostatic pressure can both influence the lipid bilayer lateral tension which may in turn further slow down the dynamics of DPPC from head to toes[31].

**NMR analysis of OmpX NMR signals in lipid nanodiscs.** OmpX from *E. coli* belongs to a large family of outer MPs that promotes bacterial adhesion to host epithelial and immune cells. Many atomic structures have been obtained by X-ray crystal-

lography and NMR spectroscopy in various media. OmpX consists of an eight-stranded antiparallel $\beta$ barrel whose strands are interconnected with three short periplasmic turns and four extracellular loops of varying lengths (Fig. 4)[32]. A plethora of NMR data of OmpX in lipid nanodiscs at atmospheric pressure is available, in particular, backbone amide proton ($^1$H$^N$) and nitrogen ($^{15}$N) and side-chain $^{13}$CH$_3$ chemical shifts (BMRB Entry 18796 (NMR structure of OmpX in phospholipid nanodiscs))[24,33] (Figs. 5a and S6b, c). Thanks to that information, we were able to follow the barotropic behavior of many $^1$H$^N$, $^{15}$N and $^{13}$CH$_3$ chemical shifts (Fig. 4) through 2D heteronuclear experiments as NMR signals undergo a continuous evolution in chemical shifts with increasing pressure (Fig. S7). The OmpX NMR signals were collected concomitantly with the previously described DMPC NMR signals (vide supra). The ramp in pressure was applied at two different temperatures, i.e., 25 °C and 40 °C. Importantly, $^1$H$^N$, $^{15}$N, and $^{13}$CH$_3$ signals of OmpX are identical for most of them or very slightly shifted at atmospheric pressure before and after the pressure ramp confirming the perfect stability of OmpX/nanodisc complexes upon pressurization and depressurization (Fig. 3b, c).

We followed the evolution under pressurization of OmpX NMR chemical shifts as they represent very sensitive parameters to detect subtle structural changes in biomolecules. In the pressure range tested (≤1500 bar at 25 °C and ≤1750 bar at 40 °C), the pressure dependence of $^1$H, $^{15}$N, and $^{13}$C chemical shifts is essentially linear (Fig. S7) and could be characterized with a first-order coefficient (slope) $b$ (in ppb/bar) for each spin observed (Fig. 4). In the case of OmpX where conformational fluctuations are largely restricted to a single conformer, $^1$H$^N$ and $^{15}$N pressure shifts both illustrate local structural modifications in the polypeptide backbone upon pressurization of the protein. In particular $^1$H$^N$ shift is predominantly associated with hydrogen bonding while $^{15}$N shift depends in addition on backbone dihedral angles[34]. Hence, $^1$H$^N$ pressure shifts are easier to interpret as a downfield/upfield shift ($b > 0/b < 0$) results from a shortening/extension and a straightening/loosening of the NH–O hydrogen bond[34].

The 62 $^1$H$^N$$b^{1H^N}$ coefficients displayed in Fig. 4 indicate a non-uniform compression of the $\beta$-barrel at both temperatures tested. Such a distortion of $\beta$-sheets has already been observed in soluble proteins[34–36]. Numerous amino acids display a negative $b^{1H^N}$ coefficient indicating the presence of local expansion of the barrel which is often counterbalanced by a strengthening of other hydrogen bonds located in the vicinity. Some largest deviations (> 2 × standard deviation, STDEV) observed concern residues located spatially close to residues whose side-chain CH$_3$ display a slow chemical exchange at either $^1$H and $^{13}$C chemical shift timescales (I132) or $^1$H chemical shift timescale only (V137, known to be involved in virulence[32]) between a major ($I$) and a minor ($II$) conformation (Fig. 5a). This concerns F107 and R131 (the latter also displays a slow chemical exchange between two

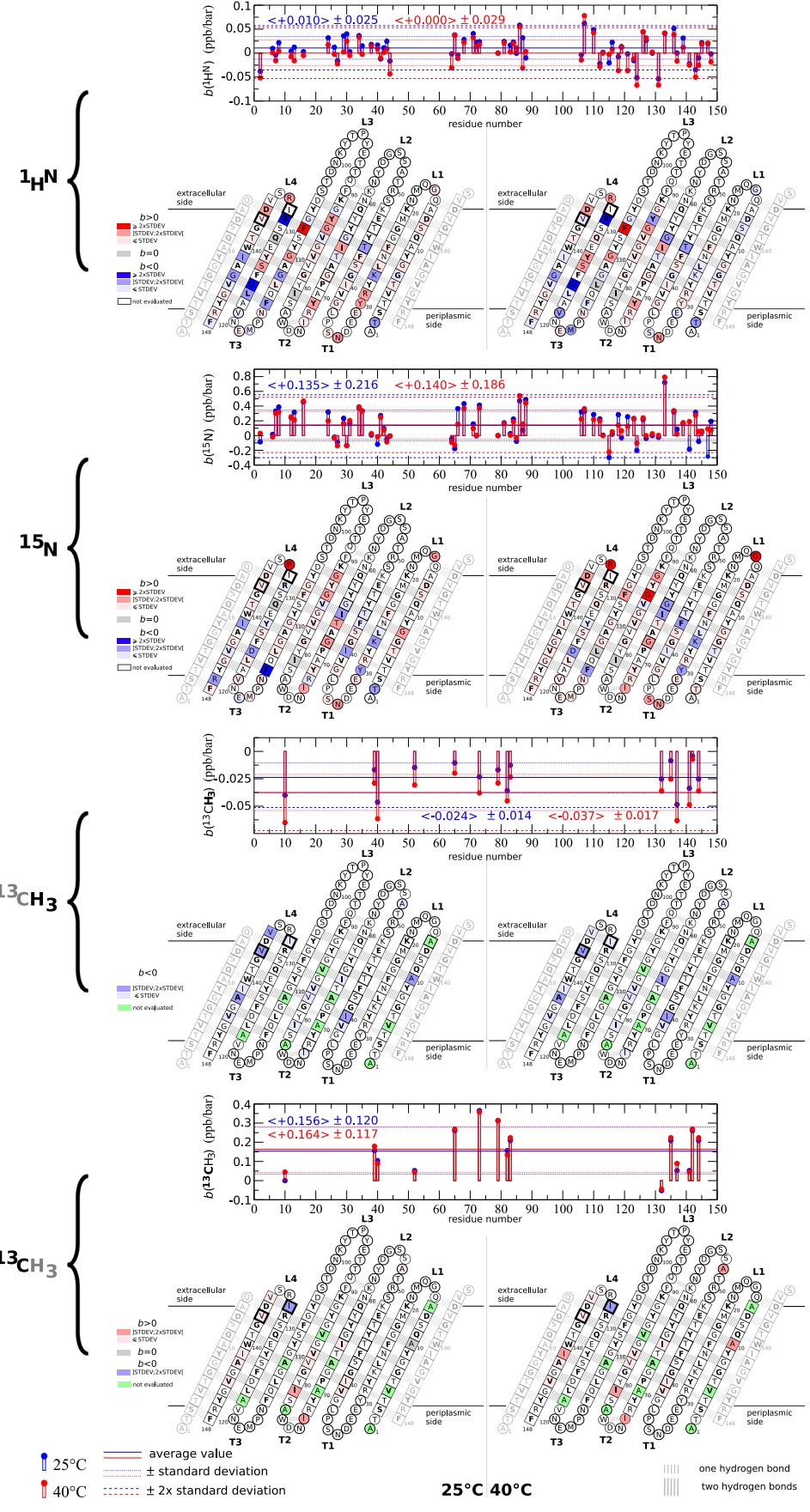

**Fig. 4 Amide $^1$H$^N$ and $^{15}$N and methyl $^{13}$CH$_3$ pressure induced chemical shift variations of OmpX.** The linear $b$ coefficient per residue is both represented on a histogram and on a snake diagram of OmpX. In the histograms, the average and standard deviation values are indicated and also represented by solid (average), dotted (±STDEV), and dashed (±2 × STDEV) lines. For residues that display a signal split in two at 25°C and 1 bar in the 2D $^1$H,$^{15}$N experiment (Fig. S6b), $b^{1H^N}$ and $b^{15N}$ have been evaluated for the most intense peak. In the snake diagrams of OmpX, residues in β-strands are shown in a square (I132 and V137, which display a split signal in 2D $^1$H,$^{13}$C experiments, with a bold square), the other in circles. Residues of the barrel that points their side chains to the lipid bilayer are written in bold letters. **L** and **T** stand for extracellular loops and periplasmic turns, respectively. Source data are provided as a Source Data file.

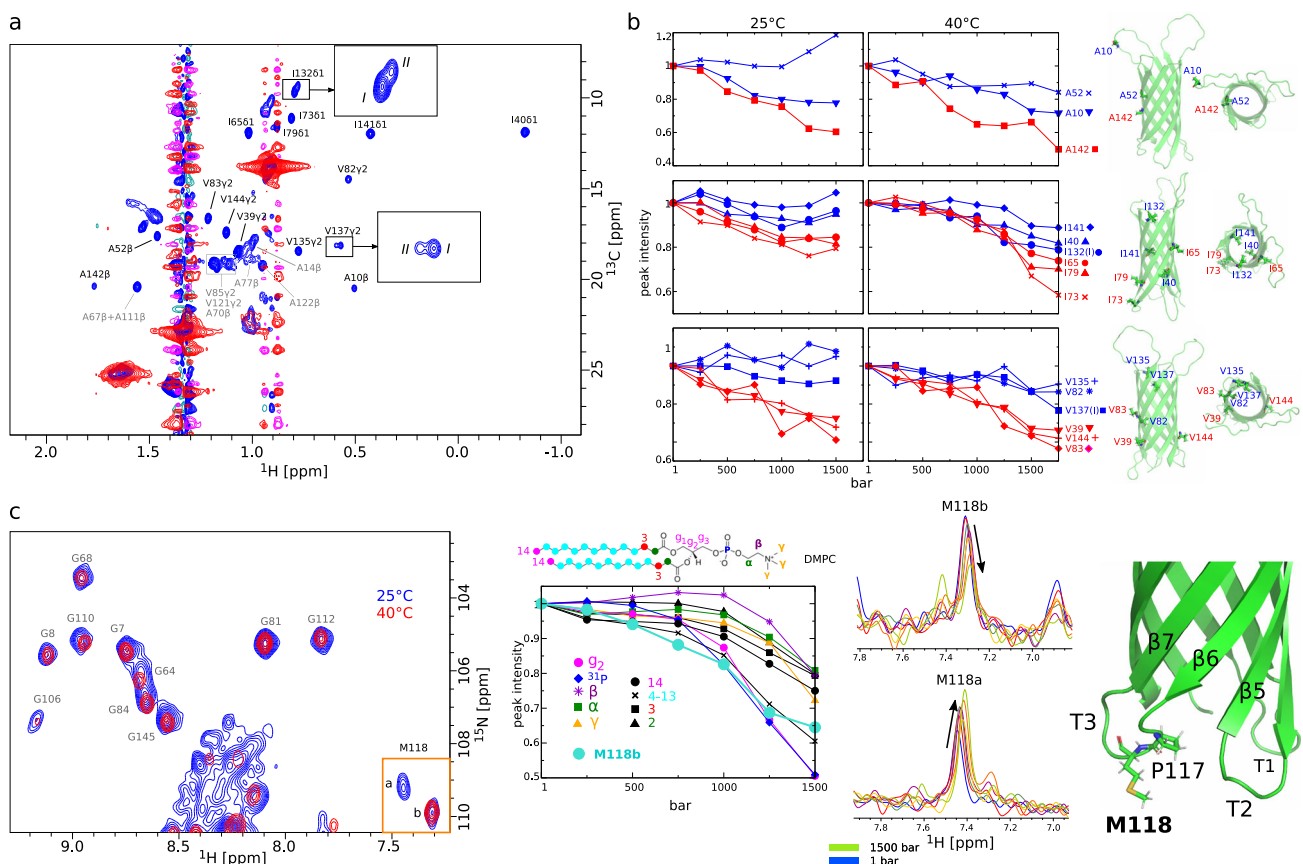

**Fig. 5 Barotropic behavior of $^{13}$CH$_3$ (a,b) and M118 amide $^1$H (c) NMR signals of OmpX. a** 2D SOFAST-HMQC spectrum (in blue) of perdeuterated OmpX with specific $^{13}$C-labeled and protonated methyl groups in Ala, Ile, ($\delta_1$), and Val (proS) residues in protonated DMPC nanodiscs (here at 40°C and atmospheric pressure). The same experiment performed with empty nanodiscs is superimposed in red. In black and gray are OmpX residues that have been taken into consideration and excluded (due to unfavorable chemical shift dispersions), respectively. The close-ups represent an enlargement of I132 and V137 signals. **b** Barotropic evolution of $^{13}$CH$_3$ relative intensities at 25°C (left) and 40°C (right) (from top to bottom, Ala, Ile, and Val residues). In red and blue have been distinguished residues whose side-chains point toward the lipid bilayer or the cavity of the barrel, respectively. Beside each panel are a membrane side and extracellular views of a cartoon representation of OmpX NMR structure in nanodiscs with Ala, Ile, and Val residues represented by sticks (pdb ID = 2M06 (NMR structure of OmpX in phospholipid nanodiscs)[24]). **c** Barotropic evolution of M118 amide proton $a$ and $b$ conformations at 25°C (see also Fig. S6b). The 2D superimposed $^1$H,$^{15}$N experiment collected at 40°C has been shifted to match the glycine signals observed at 25°C to ease the analysis of the barotropic evolution of signals $a$ and $b$ of M118 at 25°C. The diagram represents a comparison of the evolution of lipid NMR signal intensities (from Fig. 2e) with the proportion of signal of peak $b$ vs. peaks $a + b$ of M118. At 1 bar, M118b represents 52.4% of the sum of the intensities $a+b$ (normalized at 1 in the diagram) and 33.8% at 1500 bar (≡ 0.645 in the diagram). The sum of the intensities $a+b$ is constant over the pressure range at ±6%. The 1D $^1$H superimposed spectra correspond to rows for peaks $a$ and $b$ of M118 extracted from 2D $^1$H,$^{15}$N experiments collected at 25°C between 1 and 1500 bar. The cartoon on the right represents the periplasmic edge of the barrel of OmpX and shows in sticks M118 residue. Source data are provided as a Source Data file.

conformations at $^1$H$^N$ and $^{15}$N chemical shift timescales at 25 et 40 °C, Fig. S6b, c) on the extracellular side that is interlinked by a double NH–O hydrogen bond where the large compression F107(NH–O)R131 ($b^{1H^N} = +2.5$ and $+2.7 \times$ STDEV at 25 °C and 40 °C) is compensated by a large loosening of the reciprocal R131(NH–O)F107 ($b^{1H^N} = -2.2$ and $-2.3 \times$ STDEV at 25 °C and 40 °C) hydrogen bond. On the opposite side of the barrel, the largest deviations are observed for D124(NH–O)G143 ($b^{1H^N} =$

−2.1 and $−2.30 \times$ STDEV at 25 and 40 °C) and G143(NH–O) D124 ($b^{1H^N} = -1.4$ and $-1.73 \times$ STDEV at 25 °C and 40 °C) indicating that these two head to tail hydrogen bonds are both expanded upon pressurization.

Downfield and upfield $^{15}$N pressure shifts also confirm the distorsion of the barrel upon pressurization (Fig. 4) with a maximum distorsion of the peptide plane observed for the residue next to I132, i.e., residue R133 ($b^{15N} = +3.3$ and $+4.3 \times$ STDEV at

25 and 40 °C). Even though we were not able to unambiguously assign backbone amide proton and nitrogen of I132, we could follow the pressure shift of one of its methyl groups ($^{13}CH_3$–$\delta_1$–I132). On the 15 $^{13}C$ of methyl groups followed in this study, $^{13}CH_3$–$\delta_1$–I132 was the only one that displays a negative $b^{13C}$ coefficient and a significant deviation from the average ($-1.4 \times$ STDEV at both temperatures). Hence, both $^{15}N$ and $^{13}C$ pressure shifts of, respectively, R133 and I132 indicate large distorsions of hydrogen bonds and/or torsion angles at the extracellular edge of the 7th $\beta$ strand.

Globally, pressure shifts collected at 25 °C and 40 °C do not show significant differences in both average and standard deviation values (Fig. 4). This indicates that the hydrogen bond and torsion angle variations reflect mostly the impact of the hydrostatic pressure on the protein itself instead of the impact of the lipid phase. At 25 °C, but not at 40 °C, we observed at 1 bar doubled peaks in 2D $^1H^N,^{15}N$ spectra (Fig. S6b, c), especially for residues in periplasmic turns and at the C–terminal end but also at the top of the 8th $\beta$ strand, i.e., in more flexible regions close to the bilayer (Fig. S6b). Interestingly, at 25 °C, upon pressurization we observed a communicating vessel between the two well-resolved signals of M118 from a conformation that corresponds to a signal observed at 40 °C, so corresponding to the fluid phase, to another conformation whose population increase concomitantly with the gelation of the bilayer (Fig. 5c). Hence, that residue, which is connected to a proline (P117), might be a sensor of a lipid phase-dependent conformational state at that third periplasmic turn.

Comparing the evolution of the peak intensities of methyl groups at 25 °C and 40 °C, data indicates that methyl groups pointing to the bilayer show a dramatic signal reduction while methyl exposed to the interior of the $\beta$-barrel display much weaker variations (Fig. 5b). As OmpX delays the gelation process, and this is particularly pronounced at 40 °C (Fig. 2b, d), this phenomenon might not be, or not only be due to a reduction of the dynamics of lipids upon pressurization but also to variations in other physical properties of the bilayer. On the other hand, we did not observe a significant and global pressure-dependent impact on the quality of 2D $^1H$, $^{15}N$ spectra upon pressurization (Fig. S7), especially at 25 °C where lipid dynamics is impacted in the presence of OmpX at pressures above 1000 bar (Fig. 2d). This contrasts with the loss of NMR signals that have been observed for OmpX embedded in smaller DMPC nanodiscs upon gelation of the lipid bilayer by lowering the temperature at atmospheric pressure[37].

**NMR analysis of BLT2 NMR signals in lipid nanodiscs**. GPCRs represent a very large family of eukaryotic integral membrane proteins involved in a countless number of fundamental biological processes[38]. In accordance with their versatile pharmacology, these receptors display a complex and variable conformational ensemble which is not limited to a simple two-state model describing an equilibrium between an inactive and an active state. The current model of signal transduction through eukaryotic membranes by GPCRs relies on a complex conformational plasticity[39,40]. This presumably could explain the ability for the same GPCR to lead to distinct functional outcomes[41]. In addition, the activity of GPCRs can be influenced by various allosteric modulators[42], such as organic compounds, ions, or lipids. In the latter case, lipids could either bind to specific allosteric sites, e.g., ref. [25,43], topographically distinct from the orthosteric ligand binding site, or have an incidence on GPCR signaling by modulating the physical properties of the membrane[44,45]. Among these properties, the phase behavior of the surrounding lipid bilayer may play a role in the activity of

GPCRs considering these receptors and associated effectors (i) are present in various membranes displaying different lipid compositions and protein concentrations, and (ii) could co-compartmentalize in membrane sub-domains, such as lipid rafts and caveolae[46] that are enriched in glycosphingolipids and cholesterol.

Despite a limited characterized pharmacology, the BLT2 receptor has the advantage on a spectroscopic level of having only two transmembrane methionines and single isoleucine in its entire sequence. Although NMR-derived dynamic information will be restricted locally around these three residues, this opens the possibility to explore the impact of the dynamics of lipids on three strategic locations in the transmembrane region of this protein which exhibits a complex conformational space. Indeed, the conformational ensemble of BLT2 GPCR in POPC/POPG and DPPC lipid discs was further studied by two-dimensional (2D) $^1H,^{13}C$ SOFAST-HMQC NMR spectroscopy[47] between 1 and 2500 bar through three transmembrane $^{13}CH_3$ reporters, in residues M105$^{3.35}$, M197$^{5.54}$, and I229$^{6.40}$ (Ballesteros numerotation is indicated in superscript[48]; see also GPCRdb[49] BLT2 receptor homology models in Fig. S8 and methyl assignments in Fig. S9). Owing to the very high-resolution spectra offered by receptor deuteration[50] and the use of a strong magnetic field, we can observe for these three residues the co-existence of several cross-peaks reporting on the complex receptor conformational landscape (Figs. 6 and 7), as previously observed for BLT2 in DMPC nanodiscs[51]. As judged from M105$^{3.35}$, M197$^{5.54}$, and I229$^{6.40}$ methyl signals, the conformational landscape of BLT2 strongly depends on pressure, lipid composition, and the presence of the agonist (Figs. 6 and 7). At pressures ≥ 500 bar, the conformational ensemble for these three residues is easier to identify due to a better signal-to-noise ratio and more favorable chemical exchange rates (vide infra § Barotropic evolution of conformational inactive sub-states of BLT2): three and two peaks in slow chemical exchange are visible for M105 and M197, respectively, and four to five co-existing peaks are visible for I229 which is located in a more dynamic region (see below). This indicates that these three methyl NMR reporters are sensitive to receptor conformational plasticity. Importantly, pressure had also reversible effects of NMR spectra of BLT2, which indicates the receptor was not deteriorated by the pressure ramp (Fig. 3b, c).

We further analyzed in depth the evolution of the relative populations of the different species under pressure to probe the coupling between lipid dynamics and GPCR conformational landscape. This could reveal either a direct volume effect on the receptor itself such as a change in the inter-helical gap volumes upon compression or a pressure-induced modulation of lipid-receptor interactions that could be mediated by the pressure-dependent lateral compressibilities and bilayer thicknesses[31], or both.

In the absence of ligand, at atmospheric pressure, I229 in POPC/POPG nanodiscs displays several highly populated states that correspond to the initial reservoir of populations in the activation of the receptor (Figs. 6 and 7), as already observed in DMPC nanodiscs[51]. In addition to these predominant inactive (I) states, additional sparsely populated states are also visible and correspond to pre-active-like (PA) states (i.e., in the absence of a G protein)[51] (Fig. 6a), as confirmed by the increase in their intensity upon addition of full agonist leukotriene B4 (LTB4). The same observations were done on M105 and M197, with peaks C and D (see Fig. 7b) assigned to the pre-active-like states. The NMR analysis of BLT2 in nanodiscs of various compositions and in presence of agonists indicates that BLT2 preserves a complex conformational dynamic landscape in eukaryotic membranes enriched in POPC/POPG or in the more rigid DPPC membrane.

In the present study, we observed that the relative fraction of PA states for I229 appears to be negatively correlated with lipid

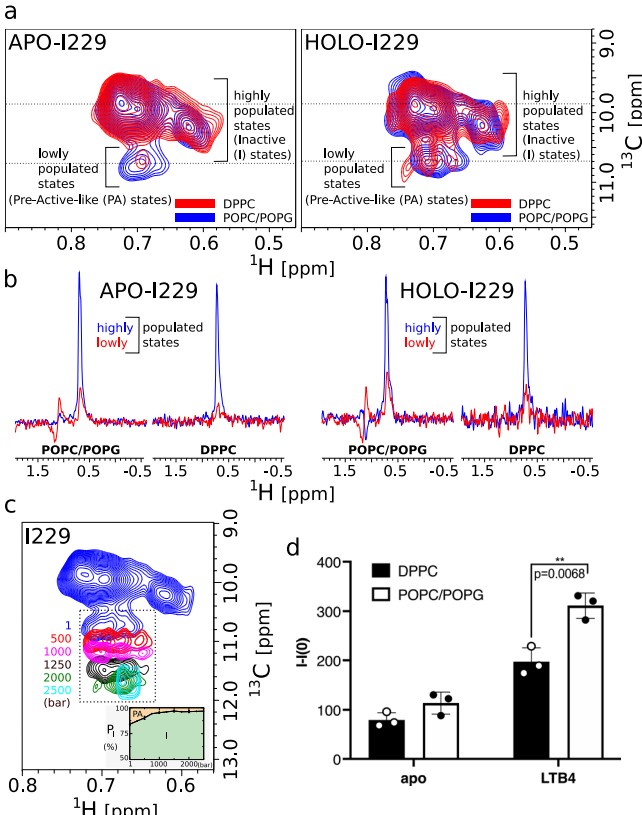

**Fig. 6 Impact of the lipid phase on BLT2 activation. a** Close-ups in the $^{13}CH_3$-$\delta_1$-I229$^{6.40}$ region of two superimposed $^1H$,$^{13}C$ SOFAST-HMQC experiments were recorded at atmospheric pressure with apo (left) and holo (right) BLT2 in POPC/POPG (in blue) or DPPC (in red) nanodiscs. The dotted lines report on $^1H$ 1D spectra displayed in **b**. Spectra were scaled to the most intense cross peak in the two spectra. **b** Extracted rows in the $^1H$ dimension from spectra displayed in **a**. The dispersive signal ~1 ppm corresponds to $T_1$ noise from the intense signal of lipid methylene protons. **c** Barotropic evolution of $^{13}CH_3$-$\delta_1$-I229$^{6.40}$ pre-active-like (*PA*) NMR signals of apo BLT2 in POPC/POPG nanodiscs. For clarity, only the lowly populated conformations are represented for pressures above 1 bar. The inset below spectra represents the relative proportion of I229 inactive (*I*) vs. *PA* states along the pressure ramp. **d** Binding of GTPγS to Gα$_i$β$\gamma$ catalyzed by BLT2-containing nanodiscs at atmospheric pressure in the absence or presence of LTB4. Data are presented as raw fluorescence intensity under a given condition (I) minus measurement under the same condition for the G protein but in the absence of any particle (I(0)) as mean ± stdev of three experiments. Statistical values were obtained by means of unpaired Student's *t* test (*0.01 < *p* < 0.05, **0.001 < *p* < 0.01). Source data are provided as a Source Data file.

stiffening, since its population is lower in the following conditions: (1) when BLT2 is embedded at atmospheric pressure in the gel-phased DPPC nanodiscs compared to the more fluid POPC/POPG nanodiscs (Fig. 6a, b)[51]; (2) when lipid stiffening is induced by pressure on BLT2 embedded in POPC/POPG and DPPC (Fig. 6c) whether the agonist is present or not (Fig. S10). In DPPC nanodiscs, *PA* signals are poorly populated in the apo or holo states compared to POPC/POPG nanodiscs and even not present at pressures greater than 250 bar, i.e., under conditions where lipids are the most rigid (Fig. S11). All these observations highly suggest that the gelation of the lipid phase destabilizes states located further along the activation pathway to the benefit of the initial reservoir of inactive populations. Nucleotide exchange assays further confirmed a significant decrease in

receptor-catalyzed $G_i$ protein activation at atmospheric pressure, both in the absence and in the presence of agonist, when the receptor was embedded in the DPPC nanodiscs compared to the POPC/POPG nanodiscs. Hence both NMR and nucleotide exchange assays demonstrated a decreased population in receptor *PA* states for BLT2 in the gel phase, and in the more rigid gel phase, DPPC compared to the more fluid POPC/POPG nanodiscs (Fig. 6d). Enhanced fluidity in the nanodisc bilayer may then favor pre-active states in BLT2.

When a protein exists in an equilibrium between different folded conformations characterized by distinct partial molar volumes, pressure tends to stabilize the conformations with low partial molar volumes[18]. When sub-state relative populations are available for a range of pressure, for example from NMR spectra, the difference in partial molar volumes ($\Delta V$) between two sub-states, or ensemble of sub-states, can be readily retrieved from the relationship $p_j/p_i = \exp[-(\Delta G^0_{i \rightarrow j} + \Delta V_{i \rightarrow j} \times P)/RT]$, where $p_i/p_j$ are the relative populations of $i$ and $j$ states, $\Delta G^0_{i \rightarrow j}$ being the Gibbs free energy in the absence of pressure, $R$ the gas constant, $T$ the absolute temperature and $P$ the pressure[52]. Doing so, the molar volume difference between I229$^{6.40}PA$ and $I$ states $\Delta V^{I229}_{PA \rightarrow I}$ was estimated to −50.6 ± 3.7 Å$^3$ and −40.0 ± 2.6 Å$^3$ with BLT2 in POPC/POPG nanodiscs in the absence or presence of the agonist, respectively (Table 1 and Fig. 8). Hence, in POPC/POPG, I229 inactive states are associated with a more compressed structure than the pre-active/active-like states, but this difference is slightly attenuated in the presence of the ligand. In DPPC, the absence of *PA* peaks at a pressure above 1 bar, unfortunately, precluded the volume analysis.

As for I229, the populations of the lowly populated states of M105 (peak $C$) and M197 (peak $D$) increase noticeably by the addition of the agonist at 1 bar (not shown) and decreased along the pressure ramp (Fig. 7b), confirming their assignment to the pre-active/active-like state and suggesting a cooperative conformational change along with the receptor upon ligand binding, as already observed with DMPC nanodiscs[51]. Therefore, M105 and M197 both report together with I229 on the destabilization of the pre-active states at high pressure, reinforcing a collective response to pressure and limited spatially restricted pressure-induced conformational changes. As observed with I229, the methyl signals of M105 and M197 were at the same chemical shifts in DPPC and POPC/POPG nanodiscs for the different structural states but at varying intensities, suggesting limited direct and specific lipid/protein interactions and revealing similar structural states in these two different lipid compositions (Fig. S14). We noted that the $\Delta V_{PA \rightarrow I}$ measured in this study for the methionines are systematically larger (10–20 Å$^3$ in absolute value) for BLT2 embedded in POPC/POPG than in DPPC nanodiscs. The current data does not allow to accurately estimate the relative contribution of lipid dynamics and direct pressure effect on the receptor conformational energy landscape, but this large volume effect of substituting DPPC by POPC/POPG firmly establishes that lipid dynamics controls, to some extent, the conformational energy landscape of BLT2. In the crystal liquid state, the lipid packing is not as tight as in the gel phase, and the resulting larger reservoir of void volumes, and hence the higher compressibility of POPC/POPG versus DPPC bilayers might contribute to the distinct conformational redistribution of BLT2 upon application of pressure.

All data pointed to the destabilization of the pre-active state at high pressure to the benefit of the inactive states. Residue I229 is close to the bottom of helix *VI* (estimated at one-third of the total length of the helix starting from the cytoplasmic end; see Fig. S8) and as such is a good reporter of the large conformational variations that are known to occur upon activation in this part of

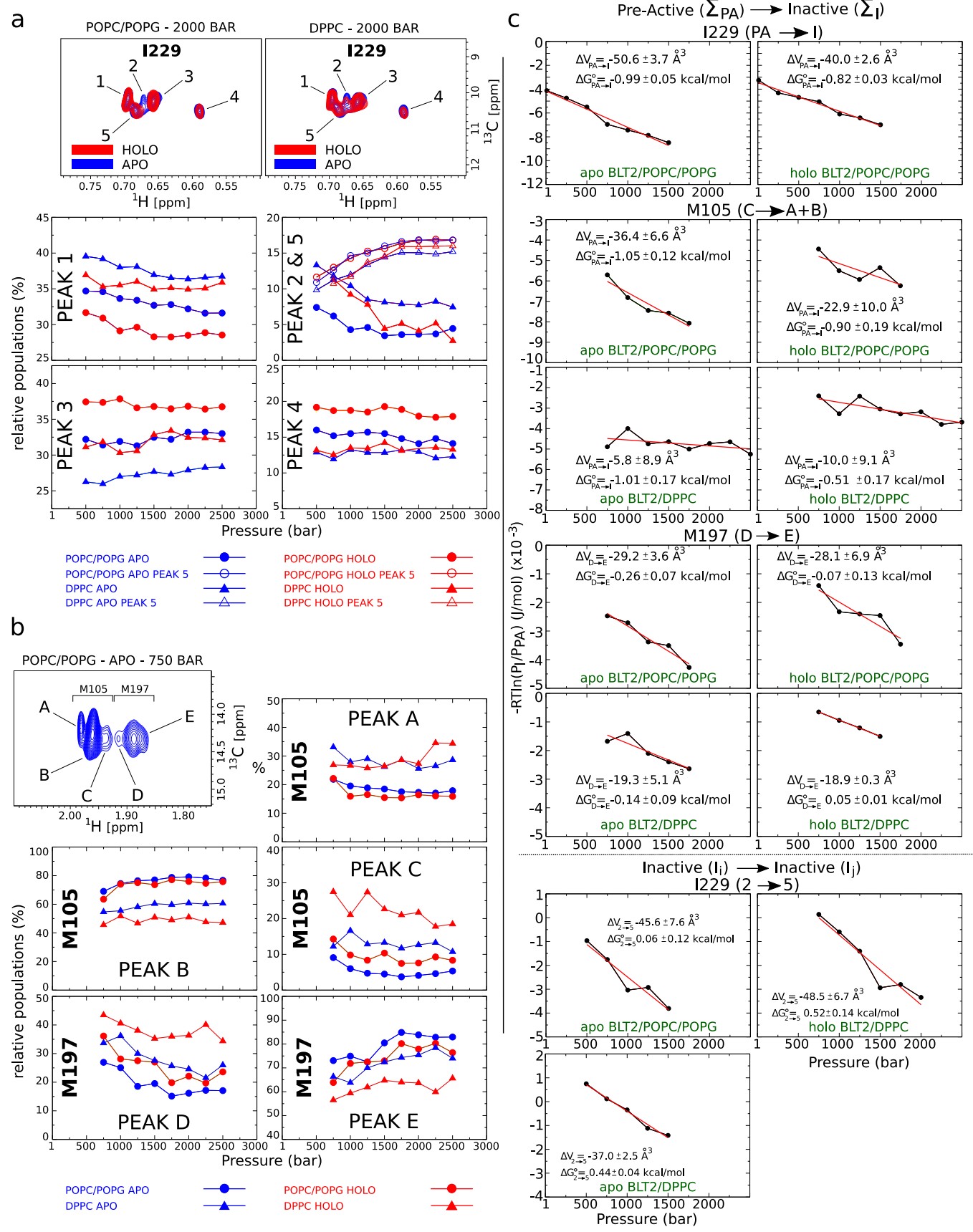

**Fig. 7 Barotropic behavior of I229, M105, and M197 $^{13}CH_3$ NMR signals. a** $^{13}CH_3$-$\delta_1$-I229 high-populated states: (top) Close-ups in the I229 region of two superimposed 2D $^1H$,$^{13}C$ spectra of BLT2 in POPC/POPG (left) or DPPC (right) nanodiscs in the absence (in blue) or in the presence (in red) of LTB4 (illustration at 2000 bar). *1* to *5* indicate the different co-existing sub-states. That figure aims at showing the difference observed for peaks *2* and *5* between the two lipid compositions (see also the graph labeled PEAK 2 & 5 below and Fig. S12 and S13). (bottom) Barotropic evolution of the populations of I229 sub-states (legend at the bottom of the figure). **b** Close-up in the $^{13}CH_3$-$\epsilon$-M105 and M197 region of a 2D $^1H$,$^{13}C$ spectrum (illustration with BLT2 in POPC/POPG nanodiscs at 750 bar; assignments are based on the M197L–BLT2 mutant, Fig. S9). The evolution of the relative populations for M105 (peaks *A*, *B*, and *C*) and M197 (peaks *D* and *E*) are indicated in the panels below. Populations of I229 were evaluated at $\geq 500$ bar and of M105 and M197 at $\geq 750$ bar, because of too many sub-states and/or a low signal-to-noise ratio at lower pressures. **c** Quantitative analysis of volume and free energy differences between the sum of the Pre-Active ($\Sigma_{PA}$) and Inactive ($\Sigma_I$) states for I229, M105, and M197 residues and between I229 *2* and *5* inactive states. Data are directly related to the populations displayed in **a** and **b** in a pressure range where variations in populations were observed. $\Delta G_{i \to j}$ and $\Delta V_{i \to j}$ extracted from the linear fit represented in red are indicated on each graph. Source data are provided as a Source Data file.

the helix with an outward movement up to 14 Å on the cytoplasmic side. This is in accordance with the larger $\Delta V_{PA \to I}$ values for I229 compared to M197 and M105 in POPC/POPG. The pressure dependence of $^{13}C$-methyl chemical shifts of I229 is not linear in the case of *PA* sub-states (Fig. 8). This means that for these I229 sub-states the compressibility varies with pressure. This nonlinearity was not observed for the initial reservoir of inactive and highly populated *I* conformers, and not for sub-states of M105 and M197 whose methyls do not point toward a cavity, including *PA* sub-states *C* and *D* (Fig. 8). For soluble and globular proteins this pressure-dependent compressibility has been ascribed for residues close to water-accessible cavities, suggesting that the nonlinear shift is related to the hydration of water-accessible cavities as increasing hydrostatic pressure then forces water molecules into protein interiors[53]. It has been observed for globular proteins that water-buried cavities can be enlarged under pressure[34]. Indeed, volume changes owing to the hydration of atoms inaccessible in the predominant low energy state and that become exposed to water under hydrostatic pressure lead to a positive contribution to the volume comparable in absolute value to the constrictions of the voids or even greater, explaining why in some cases some proteins can be stabilized by pressurization[54]. In the case of BLT2, the hydration of the cavity exposed to the cytosol is in accordance with the volume increase of 50 Å$^3$ that is measured for the transition from the inactive to pre-active states of I229.

Taken together, in BLT2, pressurization destabilizes active-like states that are sparsely populated at atmospheric pressure. This contrasts with other studies of pressurized GPCRs conducted in detergent solutions[55,56]. This result highlights the opposite response of receptors to pressure, in agreement with different void volume distributions, variation of the hydration pattern, and a distinct coupling to the surrounding lipids or surfactants.

We next analyzed the methyl signal evolution under pressurization of the highly populated inactive sub-states assigned for residues I229$^{6.40}$, M105$^{3.35}$, and M197$^{5.54}$ (Fig. 7). Surprisingly, the absolute signal intensity of these sub-states substantially increased with pressure in both nanodiscs (by a factor of 3–5 in intensity or two in volume, see Figs. 9 and S15C, E), which cannot be explained by the inactive/active state redistribution or denaturation of the protein-based of the present data. Since the hydrodynamic behavior of nanodiscs was not dramatically modified at high pressure (see the analysis of lipid translational diffusion at 1 and 2500 bar by Diffusion Ordered Spectroscopy, DOSY[57] in SI), confirming that the nanodiscs retained their molecular integrity at high pressure, we concluded that pressure-induced a major BLT2 dynamic change within nanodiscs leading to sharpened peaks, in a manner independent of the lipid composition. The same observation concerned signals of the cholesterol (1 mol%), indicating that this lipid specifically interacts with the receptor while in the absence of the protein its NMR signal intensities decrease like POPC/POPG or DPPC

**Table 1 Quantitative analysis of volume and free energy differences induced by pressurization between the sum of the pre-active ($\Sigma_{PA}$) and inactive ($\Sigma_I$) states for I229, M105, and M197 residues (Top) and between I229 *2* and *5* inactive states (Bottom) (data derived from Fig. 7) and directly related to the populations displayed in Fig. 8 in a pressure range where variations in populations were observed.**

| $\Sigma_{PA} \longrightarrow \Sigma_I$ ($\Sigma_{Pre-Active}$) 7D2($\Sigma_{Inactive}$) | $\Delta V_{i \to j}$ (Å$^3$/molecule) | $\Delta G^0_{i \to j}$ (kcal/mol) |
|---|---|---|
| *I229* (cf. Fig. 6a) | | |
| apo BLT2/POPC/POPG | −50.6 ± 3.7 | −0.99 ± 0.05 |
| holo BLT2/POPC/POPG | −40.0 ± 2.6 | −0.82 ± 0.03 |
| *M105* (*C* $\longrightarrow$ *A* + *B*) (cf. Fig. 7b) | | |
| apo BLT2/POPC/POPG | −36.4 ± 6.6 | −1.05 ± 0.12 |
| holo BLT2/POPC/POPG | −22.9 ± 10.0 | −0.90 ± 0.19 |
| apo BLT2/DPPC | −5.8 ± 8.9 | −1.01 ± 0.17 |
| holo BLT2/DPPC | −10.1 ± 9.1 | −0.51 ± 0.17 |
| *M197* (*D* $\longrightarrow$ *E*) (cf. Fig. 7b) | | |
| apo BLT2/POPC/POPG | −29.2 ± 3.6 | −0.26 ± 0.07 |
| holo BLT2/POPC/POPG | −28.1 ± 6.9 | −0.07 ± 0.13 |
| apo BLT2/DPPC | −19.3 ± 5.1 | −0.14 ± 0.09 |
| holo BLT2/DPPC | −18.9 ± 0.3 | +0.05 ± 0.01 |
| **Inactive(*i*) $\longrightarrow$ Inactive(*j*)** | $\Delta V_{i \to j}$ (Å$^3$/molecule) | $\Delta G^0_{i \to j}$ (kcal/mol) |
| *I229* (*2* $\longrightarrow$ *5*) (cf. Fig. 7a) | | |
| apo BLT2/POPC/POPG | −45.6 ± 7.6 | +0.06 ± 0.12 |
| apo BLT2/DPPC | −37.0 ± 2.5 | +0.44 ± 0.04 |
| holo BLT2/DPPC | −48.5 ± 6.7 | +0.52 ± 0.14 |

No data are available with holo BLT2 in POPC/POPG nanodiscs for transition *2*→*5* because conformation *2* disappears in the presence of LTB4 (see Figs. S12 and S13).

upon the pressure-induced gelation of the bilayer (Fig. S15). Hence, the pressurization of a nanodisc can represent an interesting way to detect specific MP lipid cofactors. The enhanced quality of the NMR spectra by applying pressure can greatly facilitate the analysis of the conformational landscape at a higher pressure and indicates that pressure application might represent a convenient tool to improve NMR studies of challenging membrane proteins by tuning lipid and/or protein dynamics.

We observed that the distribution of populations within inactive states of BLT2 was largely insensitive to pressure (Fig. 7) but varied with agonist binding and lipid composition. Nevertheless, data shows stabilization of peak *5* at the expense of peak *2*, both in holo and apostates and in both lipid compositions (Fig. 7). Notably, we can conclude that state *5* is the more compact state and state *2* the less compact one, whereas states *1*, *3*, and *4* are of similar molar partial volumes. In addition, lipid composition, but also agonist binding, led

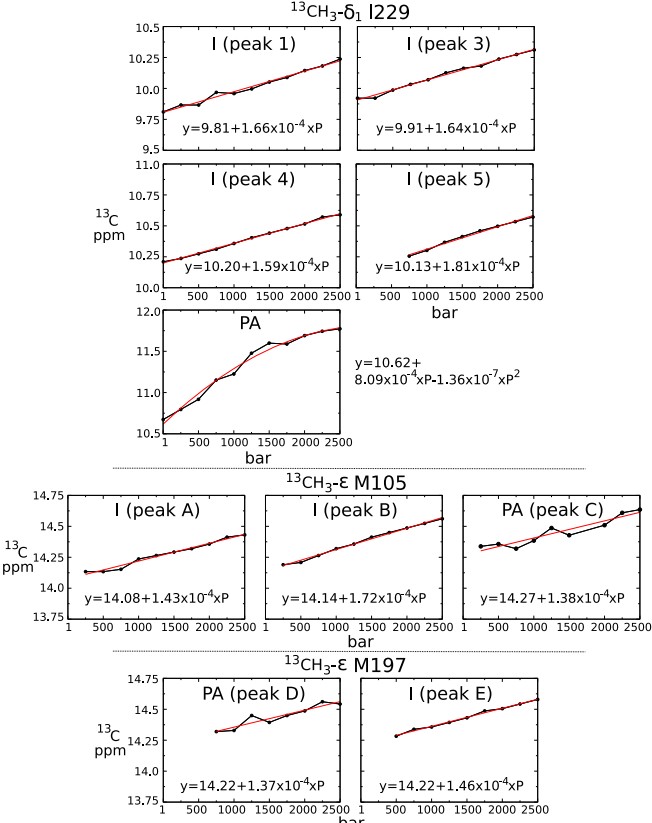

**Fig. 8 $^{13}$C chemical shift pressure-induced evolutions.** Barotropic behavior of $\delta_1$-I229, $\epsilon$-M105, and $\epsilon$-M197 $^{13}$C NMR chemical shifts of *apo* BLT2 in POPC/POPG nanodiscs. Source data are provided as a Source Data file.

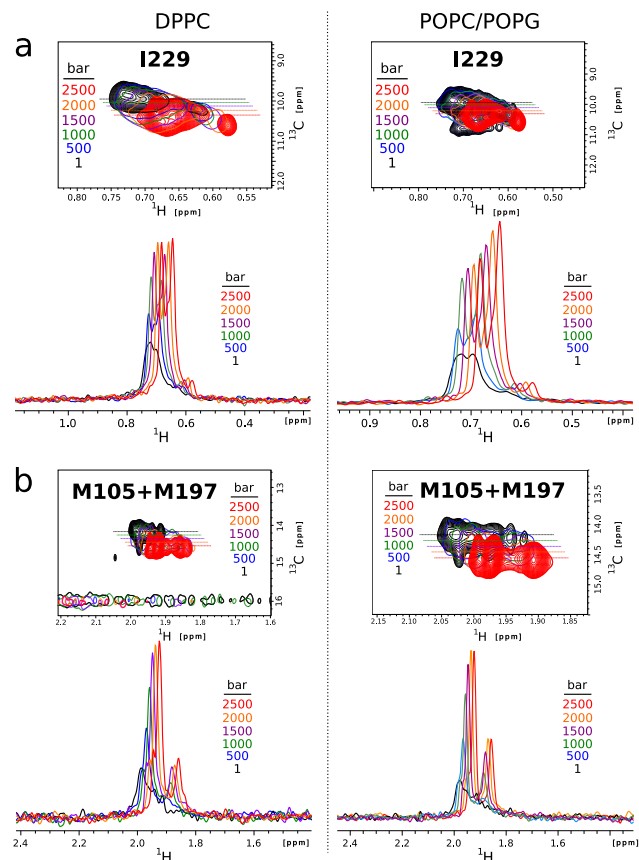

**Fig. 9 Boost in receptor methyl signal intensities upon pressurization.** Evolution of I229 (**a**), M105, and M197 (**b**) high-populated sub-state $^{13}$CH$_3$ NMR signals along the pressure ramp of the apo receptor in DPPC (left) and POPC/POPG (right) nanodiscs. **a** Close-up in the $\delta_1$-$^{13}$CH$_3$-I229[6.40] region of six superimposed two-dimensional $^1$H,$^{13}$C-SOFAST-HMQC (950 MHz $^1$H Larmor frequency) NMR spectra. For clarity, only one contour line was plotted for the experiments recorded at 500, 1000, 1500, and 2000 bar, and in the case of BLT2 in POPC/POPG nanodiscs, the low-populated states were only represented at 1 bar. **b** Same as **a** but in the $\epsilon$-$^{13}$CH$_3$-M105[3.35] & M197[5.54] region. In both **a** and **b**, the horizontal dotted lines indicate $^{13}$C frequencies at which rows along the proton dimension have been extracted and represented below each 2D superimposed spectra. In the I229 region, the rows correspond to the $^{13}$C resonance of peak 3, and in the M105+M197 region, the rows correspond to peak B (see peak labeling in Fig. 7).

to a change in $\Delta V_{2\to5}$ (see Fig. 7c) further highlighting how lipid composition impacts the pressure effect on the embedded protein. This analysis revealed here how lipid composition and/or dynamics and ligand binding can differentially redistribute void volumes between inactive states.

Thanks to the perfect resistance and reversibility of lipid nanodisc structure upon the application of pressure and the aptitude of these nanometric lipid bilayers to sample fluid-to-gel phase transitions equivalent to the ones experienced by larger lipid systems, the association of high-pressure NMR with nanodiscs represents an efficient tool to address at the atomic scale the close dynamic relationships between lipids and MPs. The limited size of the nanodiscs allows to properly judge the impact of the membrane protein on the dynamics of the neighboring lipids and two opposite effects were observed, depending on the physical properties of the lipid bilayer and the type of MP inserted. Pressure as a thermodynamic variable has the advantage to induce only volume variations, by contrast with the temperature that induces both volume and thermal energy variations which can complicate the thermodynamic analysis. In the context of lipid bilayers, pressure has also the great advantage to be compatible with high-resolution solution-state NMR spectroscopy to studying phase transition with $T_m$ below 0 °C.

We studied here membrane proteins from non-piezophilic organisms at high pressure, i.e., out of their normal functioning conditions. Immersing biomolecules under non-native environments allows understanding of the impact of environmental stress or adaptation. We show that the two used proteins tend to maintain lipid fluidity in their direct neighborhood under pressure. Although additional data will be required for confirmation, this could reveal a mechanism by which MP may keep

functioning under conditions where MP-free lipids should be rigid, as an adaptation to high-pressure environments.

We were also able to follow with great accuracy the barotropic evolution of membrane protein conformational landscapes. We observed that the sub-state redistribution can be either sensitive to lipid dynamics or not. The elastic compression of the bilayer not only modifies lipid dynamics but also the bilayer thickness which has been shown recently to have a noticeable impact on GPCR function[58]. Hence, the strategy described herein offers a new way to better understand the diversity in lipid-protein dynamic interactions and by extension of biological membranes.

## Methods

**Site-directed mutagenesis**. All the NMR experiments were carried out with a single mutant of OmpX (H100N)[32] and a triple mutant of human BLT2 receptor where the three extra-membrane methionines (M1(Nter), M325(Cter), and M349(Cter)) were mutated to alanines in order to observe the two remaining transmembrane methionines M105 and M197. An additional mutant was prepared to assign the two transmembrane methionines, M197L–BLT2. All these mutations

were introduced in the wild-type BLT2 receptor by PCR-mediated mutagenesis using the QuikChange multisite-directed mutagenesis kit (Stratagene) and the wild-type BLT2 construct as a template. Mutations were confirmed by nucleotide sequencing.

**Overexpression, purification, and reconstitution in lipid nanodiscs of OmpX and BLT2.** Bacterial expression and purification of OmpX and BLT2 were carried out as previously described in ref. [32] and ref. [50,51], respectively. The isotope-labeled proteins were uniformly deuterated, [15]N-labeled, [13]C at natural abundance. BLT2 was specifically labeled with [13]C and protonated methyls at Met and Ile residues (u-[2]H,[15]N Ile-[$\delta_1$–[13]CH$_3$], Met-[$\epsilon$–[13]CH$_3$]-BLT2) following a procedure described in[51] and OmpX was specifically [13]CH$_3$-labeled at Ala, Val (proS) and Ile ($\delta_1$-Ile) residues (TLAM kit from NMRBio). The nanodiscs were formed by adding the lipoprotein MSP1D1[59] to detergent-solubilized DMPC (1,2-dimyristoyl-sn-glycero-3-phosphocholine), or DPPC, (1,2-dipalmitoyl-sn-glycero-3-phosphocholine) or a mixture of POPC (1-palmitoyl-2-oleoyl-glycero-3-phosphocholine) and POPG (1-Palmitoyl-2-oleoyl-sn- glycero-3-phospho-glycerol) (3/2 molar ratio of POPC/POPG in the original mixture before reconstitution and confirmed by 1D [31]P NMR spectra performed on nanodiscs; Fig. S5a) phospholipids in the presence of 1 mol% of cholesterol. The reconstitution in lipid nanodiscs of OmpX and BLT2 was carried out as previously described in[24] and[50,51], respectively. Based on the mean surface area per lipid in nanodiscs of 52 Å$^2$ for DPPC, 57 Å$^2$ for DMPC, and 69 Å$^2$ for both POPC and POPG, a rough estimation of the number of lipid layers around OmpX and BLT2 is comprised between 2 and 3, considering a cross-sectional area of ~850 Å for the $\beta$-barrel[24] and of 140 Å$^2$ per transmembrane helix for BLT2 and a total bilayer area of 4400 Å$^2$ using the lipoprotein MSP1D1[59]. In the case of BLT2, the same estimation can be made based on a cross-section of rhodopsin from a molecular dynamics model in a POPC phospholipid bilayer[60]. In the absence of MP, MSP1D1 nanodiscs would contain roughly ~85, ~77, and ~64 DPPC, DMPC, and POPC+POPG lipids per leaflet, respectively, and ~67 DPPC and ~50 POPC+POPG molecules in the presence of BLT2 and 62 DMPC per leaflet in the presence of OmpX.

**The nucleotide exchange assay.** The nucleotide exchange assay using the purified G$\alpha_{i2}$ subunit was carried out as described in[61]. In brief, the rate of GTP$\gamma$S binding was determined by monitoring the relative increase in the intrinsic fluorescence ($\lambda_{exc}$ = 300 nm, $\lambda_{em}$ = 345 nm) of the G protein trimer (500 nM) in the absence or in the presence of BLT2-containing discs (100 nM) in a buffer containing 10 mM MOPS, pH 7.2, 130 mM NaCl, and 2 mM MgCl$_2$ for 40 min at 15 °C after the addition of 10 μM GTP$\gamma$S. Agonist-dependent activation was measured under the same conditions in the presence of 10 μM LTB4 (5S,12R-dihydroxy-6Z,8E,10E, 14Z-eicosatetraenoic acid).

**DLS and SEC experiments.** DLS experiments were performed with a Zetasizer Nano S instrument (Malvern). The parameters used with the Zetasizer Software were set to water for the dispersant, 1.33 for the dispersant refractive index, 1.45 for the material refractive index, 0.001 for the material absorption, and 0.8872 centipoise for the dynamic viscosity. The angle of measurement was set to 173° in order to avoid multiple scattering. SEC was performed at room temperature (RT) on an Äkta purifier 10 system (GE Healthcare) equipped with a Superose 12 10/300 GL column (void volume: 8 mL, total volume: 24 mL). The column was equilibrated with NMR buffer and 100 μL of the nanodisc samples in NMR buffer was injected the column with detection at 280 nm. The elution flow rate was set at 0.5 ml/min. For the three lipid compositions in the present study, the samples were split into two-part before performing DLS and SEC experiments at RT. After the first set of experiments collected at atmospheric pressure, empty nanodiscs were incubated at RT for 48 h followed by a pressure jump at 2500 bar (the pressure was increased by an increment of 500 bar from 1 bar to 2500 bar within ~3–4 min.) during 25 min before returning at 1 bar within ~2 min to repeat DLS and SEC experiments.

**NMR sample preparations.** For both MPs, the NMR buffer solution was 25 mM Tris-HCl (pH 8 with BLT2 and 7.3 with OmpX), 50 mM NaCl, 2 mM EDTA in 90% H$_2$O/10% D$_2$O. BLT2-containing lipid disc concentrations were comprised between 100 and ~300 μM, except for the unlabeled samples (~30–60 μM). OmpX-containing nanodisc concentrations were comprised between ~300 and ~500 μM. In all, 100 μg of LTB4 supplied in ethanol solution were used by BLT2 NMR sample. The ethanol was evaporated using a rotary evaporator and the ligand was directly dissolved by the NMR sample containing the receptor embedded in lipid nanodiscs. The proper evaporation of ethanol was confirmed by NMR. Considering the relatively high concentrations of receptor, it was not possible to work at saturation of ligand over the receptor, because of the limited solubility in aqueous solutions of LTB4 estimated ~0.015 g/L ($\equiv$ 45 μM; partition coefficient logP(octanol/water) = 5.46 estimated by the software ALOGP)[62]. This gives rise to a ligand/receptor molar ratio at most ~1/2. So, the term holo in this study corresponds to partially liganded receptors. For both lipid compositions, the difference in populations for the different sub-states of I229, M105, and M197 observed between the absence and presence of the ligand is constant over the pressure range tested (Fig. 7) suggesting that ligand solubility or affinity for the receptor seem constant between 1 and 2500 bar.

**NMR spectroscopy.** One-dimensional [1]H water-suppressed, two-dimensional [1]H,[13]C SOFAST methyl-HMQC-TROSY[47], [1]H,[15]N SOFAST-HMQC[63], and DOSY[57] experiments were performed at 950 MHz with a TCI cryoprobe. One-dimensional [31]P experiments were collected on a 600 MHz spectrometer ($\equiv$243 MHz [31]P Larmor frequency) with a TBI probe (one-dimensional [1]H water-suppressed experiments were also performed at 600 MHz with the same probe). All the NMR experiments with BLT2 were conducted at 25 °C and at 25 and 40 °C with OmpX on Bruker Avance III NMR spectrometers. 2D [1]H,[13]C SOFAST methyl-HMQC-TROSY spectra were acquired with a 200 ms recycling delay, 100 ms acquisition time ($t_{2max}$) in the direct dimension, and 12.5 ms ($t_{1max}$) in the indirect dimension. The variable flip angle for the PC9 shape pulse was set to 120 °C. The number of acquisitions per increment was 24, for a total experiment time of 29 min. 2D [1]H,[15]N SOFAST-HMQC spectra were collected with a 250 ms recycling delay, 80 ms acquisition time ($t_{2max}$) in the direct dimension, and 38 ms ($t_{1max}$) in the indirect dimension. The number of acquisitions per increment was 16, for a total experiment time of 25 min. One-dimensional [31]P experiments were acquired with 256–1024 acquisitions (experiment time comprised between ~10 and 40 min). Data displayed in Figs. 2, 7a–c, 8, and S15c–f corresponds to the mean of 2–3 repeated NMR experiments (deviations from the mean can be found in the Source Data file). Unless specified, data processing and analysis were performed with TopSpin (Bruker BioSpin) and CCPNMR[64] softwares.

The pressure apparatus used was an Xtreme-60 Syringe Pump from Daedalus Innovations LLC (https://daedalusinnovations.com/high-pressure-nmr/) allowing to reach up to 4000 bar and designed to handle incompressible fluids such as water. The NMR tube (Daedalus Innovations LLC) is made of alumina-toughened zirconia, capable of supporting 2500 bar. The outer diameter is 5 mm. The inner diameter is 3 mm for a sample volume of 300 μL. A layer of mineral oils was used as a barrier between the sample and the pressurizing medium as described in ref. [65]. Before NMR data collection at high pressure, a rapid pressure jump to 2500 bar was done outside the spectrometer to check the absence of leakage. For all experiments with BLT2 and OmpX, the pressure ramp was from 1 to 2500 bar (BLT2) and 1–1750 bar (OmpX), each 250 bar, and back to 1 bar by step of 500 bar. NMR data were collected at all steps. Additional 100 bar increments were included to precisely measure the phase transition through 1D [1]H and 1D [31]P spectra at 600 MHz. A delay of 15 min was imposed after each pressure change and prior to NMR data collection to allow system equilibration. One-dimensional [1]H spectra collected before and after 2D data collection were identical indicating equilibrium was indeed reached. The total NMR duration was typically about 24 hours for a complete pressure cycle.

**Reporting summary.** Further information on research design is available in the Nature Research Reporting Summary linked to this article.

## Data availability

Additional data supporting the findings of this work is available as Supplementary Information and Source Data files. Raw data are available from the corresponding authors upon reasonable request. Source data are provided with this paper.

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

## Acknowledgements

We are grateful to Christina Sizun and Anthony Potenza for assistance with the DLS and pressure apparatus, respectively. This work was supported by the Centre National de la Recherche Scientifique (CNRS), Université de Paris and Université de Montpellier, the Agence Nationale de la Recherche (ANR-17-CE11-0011), Laboratoire d'Excellence (LabEx) DYNAMO (ANR-11-LABX-0011) and Equipements d'Excellence (EQUIPEX) CACSICE (ANR-11-EQPX-0008) from the French Ministry of Research, and a post-doctoral fellowship Marie Curie Global Fellowship for M.C. (AlloGPCR-799376). Financial support from the IR INFRANALYTICS FR2054 CNRS for conducting the research is gratefully acknowledged.

## Author contributions

J.-L.B., E.L., and L.J.C. designed research; A.P., F.G., Q.C., M.C., E.P., M.D., C.L.B., K.M., J.-L.B., E.L., and L.J.C. performed research; A.P., F.G., J.-L.B., E.L., and L.J.C. analyzed data; and E.L. and L.J.C. wrote the paper.

## Competing interests

The authors declare no competing interest.

## Additional information

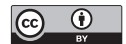

