## [Peer Review File · Nature Communications]

REVIEWER COMMENTS

Reviewer #1 (Remarks to the Author):

The manuscript by Pozza et al entitled "Exploration of the dynamic interplay between lipids....." describes the use of hydrostatic pressure as a tool to investigate lipid bilayer phase properties in empty as well as protein-occupied nanodisc membrane mimetics. The GPCR BLT2 is used as an example of a protein cargo while nanodiscs with two types of lipids - saturated DPPC vs unsaturated/charged POPC/POPG - are investigated. Effects of lipid and protein on each other under pressure are investigated and for apo and holo receptor interpreted as changes to the conformational landscape and redistribution of void volumes. The authors use predominantly NMR spectroscopy as method of investigation and propose the generality of the methodology to investigate lipid effects on proteins.

General:

1) This is an interesting and carefully conducted study which thanks to NMR spectroscopy opens a wider range of applications to the biophysical exploration of lipid bilayers at atomic resolution. However, the physiological relevance of the range of pressures used in the study is not clearly established. This should be explained.

2) While the authors promote the presence of just the three ¹³C NMR probes in BLT2 as an asset to their study (BLT2 has only 2 Met and 1 Ile), I can't help thinking that introducing this promising technology would have made a more convincing case if conducted on a well-characterised MP such as e.g. OmpX or similar. The complete assignments available in the literature would have allowed a much more global and subtle mapping of the effects investigated. As such the paper remains a mixture of introducing a promising technology while illustrating it through the use of a less than convincing target where only a low resolution view of this multispan protein is offered. Under the given circumstances we don't know for example what effect pressure and lipid choice exert e.g. on intra/extra membrane regions including the orthosteric binding pocket. The authors emphasize the ability to discriminate between multiple conformations for each of the three probes as a result of using highly deuterated BLT2 receptor, but this is beyond the point and the real issue is that only very little of the protein is covered while there remain vast 'dark regions' over which we don't know how they respond to pressure. As the authors frequently refer to the important phenomenon of void volume redistribution this is important. I do fully understand that the authors wanted to increase the impact and timeliness of the manuscript by using a GPCR but I think that the latter circumstances due to the sparse coverage not only take away from this otherwise carefully conducted investigation but also introduce an element of ambiguity.

3) Naturally it is difficult to assess the impact related to the experimental conditions where hydrostatic pressure acts omnidirectional, while a physiological membrane is predominantly exposed to lateral pressure effects and changes across the bilayer. We don't know how this fundamental difference in how pressure is applied manifests itself on lipids and protein, and the authors should add this as a caveat in their discussion.

Specific:

4) p4, 71. The authors claim that the presence of only 2 methionine and isoleucine allows the exploration of the receptor conformational landscape with great accuracy (see above). This is misleading as the presence of only 3 active NMR probes provides the receptor landscape at very low resolution. This needs rephrasing to take this into account.

5) p8, 181: The authors state that the effect of the receptor under pressure parallels the effect of cholesterol. While this is the case for DPPC, however, it seems to achieve the opposite effect for the POPC/POPG system. The current statement gives the impression that both bilayers behave in analogy as if cholesterol was added.

6) p12, 313 paragraph: The authors interpret the changes in volume between substates in a manner that suggests that agonist binding leads to a substantial void volume redistribution. In this context the appearance of additional cavities in the pre-active state needs to be commented on while taking into account more recent indications in the literature that such cavities are likely to be filled with water molecules.

7) p13, 352. Previous pressure-dependent studies were conducted on detergent solubilised membrane proteins. The authors comment briefly at the comparison of pressure dependent studies in lipid vs detergent, however, their comment is based on the comparison of different receptors. In view of the modality in the energy landscape of different receptors such a comparison will only hold when conducted on the same receptor. Please take this into account and reformulate this part of your discussion.

8) The dramatic increases in signal intensity observed on the receptor side as shown e.g. in Figure 3 are rather extraordinary and their cause needs further assessment. While this is potentially interesting and could formulate a tool for receptor studies (the authors mention this in the manuscript) just based on BLT2 there is no indication that other receptors would behave similarly.

9) Limited agonist solubility lead to a situation where due to a small amount of ligand the receptor is only partly bound to ligand. How are the ligand equilibria affected by pressure and how does the water/lipid partitioning of the hydrophobic ligand change upon application of pressure? If changes in the ligand-receptor affinity result from the application of pressure then this will lead to changes in the observed signal intensities and question the current interpretation of the spectra.

Reviewer #2 (Remarks to the Author):

In their manuscript entitled "Exploration of the dynamic interplay between lipids and membrane proteins by hydrostatic pressure" Pozza et al describe the influence of hydrostatic pressure on the phase transition behavior of empty lipid bilayer nanodiscs and containing a GPCR in complex with different ligands. The paper addresses the important question of how a membrane protein can influence the phase behavior of the surrounding lipid bilayer environment and vice versa. Since unsaturated native-like lipids show phase transition temperatures that are below 0°C, the use of pressure is an elegant way to capture the properties of the lipid bilayer using NMR spectroscopy performed at ambient temperatures.

I have a few comments and questions that should be considered by the authors:

1.) Integrity of nanodiscs after application of high pressure

Here mostly NMR 1H spectra were used as well as DOSY spectra using the lipid signals as a readout. I am wondering whether other methods have been used to confirm the presented conclusion since I am not sure if NMR is the best method to convincingly show the integrity of nanodiscs. Did the authors try size exclusion chromatography or light scattering experiments (DLS)?

2.) Phase transition of empty nanodiscs

The authors mention that DPPC nanodiscs are in the gel phase at any pressure at the chosen temperature of the NMR experiments (25°C). In order to validate the presented NMR method, it would be interesting to see whether the phase transition of DPPC can be captured if the NMR experiments are conducted at a temperature where DPPC is in the liquid crystalline phase (> 40°C). Lipid nanodiscs are stable enough for such an experiment. Legend to Figure 1: how were the error bars obtained?

The MSP protein that is part of lipid nanodisc particles may also change the phase transition behavior of the lipids. Thus, it would be helpful to compare the presented nanodisc data with liposome preparations. I fully understand the motivation of using nanodiscs for this study since GPCR signals are analyzed here as well. However, for the phase transition behavior studies, a pure lipid system would be an important benchmark.

3.) Receptor loaded nanodiscs

Since the GPCR (monomer) is relatively large and the authors apparently used MSP1D1 nanodiscs of 10 nm in diameter I am wondering how many lipids are actually present in the loaded nanodiscs and whether the observed effects (in fig. 2) could be caused by a varying degree in lipid content. If considering the surface area of the two lipids used here, it is likely that less POPC is present in the GPCR-containing nanodiscs than DPPC, thus maximizing the effect of the membrane protein.

4.) NMR spectra of the GPCR

How can the authors exclude the occurrence of partially unfolded or misfolded species of the GPCR preparation that might also contribute separate signals in the ¹³C isoleucine or methionine NMR spectra and potentially might respond to changes in pressure?

Could the higher intensity of the NMR signals at high pressure also be explained by a better homogeneity or even the presence of smaller nanodiscs?

Fig 3B: Why does the lower-field signal at ~1 ppm the 1D spectrum in red show a dispersive line shape or is even negative (blue spectrum on the right)?

5.) The authors should explain in more detail what novel insights are presented in the manuscript since many aspects of lipid phase transition behavior and the effect of incorporated membrane proteins have been explored already.

6.) Minor point: The large number of supporting information figures renders this manuscript quite hard to read. Thus, the authors might consider including some of the data into the main figures or combine data.

POINT-BY-POINT ANSWER TO REVIEWER COMMENTS

First of all, we would like to thank the two expert reviewers for having read in depth our manuscript, and for providing helpful and insightful comments that helped us to significantly increase the quality of the work. In particular, the proposal to extend the scope of the paper to other membrane proteins like the beta-barrel OmpX further expands the readership and demonstrate that the approach can be generalized to at least a second well studied membrane protein. In addition, we also demonstrate that DPPC in nanodiscs can experience fluid-to-gel phase transition upon pressurization and we confirm NMR data with Dynamic Light Scattering and size exclusion chromatography measurements to indicate that lipid nanodiscs do resist to high pressure.

Our manuscript has been deeply reworked and now includes a new series of experiments to reinforce the interest of associating lipid nanodiscs with variation in pressure instead of temperature. Below is detailed how we addressed all the points the reviewers raised. These include specific comments and additional experiments. To keep a reasonable manuscript size despite the additional experiments performed with OmpX, we shortened some parts in the analysis of pressure-induced evolution of BLT2 conformational landscape (crossed lines 519-532, 541-548, 641-674).

For convenience, in the manuscript, the modification or addition of text are written in red and parts of the text that have been conserved but moved are written in blue. The text that has been removed is written in black and crossed out. In addition to the edited manuscript, we provide a version where the modifications are not indicated to make the reading more easier.

REVIEWER COMMENTS

Reviewer #1 (Remarks to the Author):

The manuscript by Pozza et al entitled “Exploration of the dynamic interplay between lipids.....” describes the use of hydrostatic pressure as a tool to investigate lipid bilayer phase properties in empty as well as protein-occupied nanodisc membrane mimetics. The GPCR BLT2 is used as an example of a protein cargo while nanodiscs with two types of lipids - saturated DPPC vs unsaturated/charged POPC/POPG - are investigated. Effects of lipid and protein on each other under pressure are investigated and for apo and holo receptor interpreted as changes to the conformational landscape and redistribution of void volumes. The authors use predominantly NMR spectroscopy as method of investigation and propose the generality of the methodology to investigate lipid effects on proteins.

General:

1) This is an interesting and carefully conducted study which thanks to NMR spectroscopy opens a wider range of applications to the biophysical exploration of lipid bilayers at atomic resolution. However, the physiological relevance of the range of pressures used in the study is not clearly established. This should be explained.

Answer: to better situate the reader concerning the physiological relevance of the pressure range used in our study, we added few sentences in the introduction on the so-called deep biosphere (deep sea and subsurface terrestrial habitats) [lines 32-34 and 55-58, accompanied by one additional reference numbered 21]. To date, some piezoresistant organisms have been found to grow up to 1250 bar which more or less corresponds to the order of magnitude of the pressure range used in our study. Beside these physiological considerations, we also added a sentence (associated with Fig. 3) to indicate that, at the molecular level, the proteins studied in the present manuscript have most of their structures unperturbed in the pressure range we used [for OmpX this corresponds to lines 286-289 and for BLT2 this was already mentioned in the previous version of the manuscript, now on lines 455-457].

Not mentioned in the manuscript, we would like to add the following comment concerning the physiological relevance of our study. Countless studies at the atomic scale or at the single molecular level are conducted at very low temperatures (cryoEM, X-Ray, ...), in crystals, with extensive mutations (e.g. to thermostabilize GPCRs) or very high salt concentrations (histones, ...) as required to obtain nice samples or usable experimental data, but this does not necessarily mean that these studies are not valid. We believe that pressure offers a much milder modification than crystallization, or extreme temperatures, notably here since the proteins remain largely folded in the tested pressure range to retrieve structural/dynamic data (ΔV). As exemplified here, high

pressure seems also to give a dramatic improvement of the signal-to-noise ratio in NMR spectra, which offers future unique perspective to probe membrane protein (GPCRs) dynamics, at least for BLT2. We are not aware of another technique to get such spectral improvement (beyond increasing field strength and deuteration). However it is needless to say that the obtained data need to be validated by orthogonal approaches, as any new methods.

Finally, the vast majority of the biophysical studies on membrane proteins are performed in detergent solutions, and this is not the case of our study conducted with MPs embedded in a bilayer-like environment thanks to nanodiscs which represent a considerable technological advance for the community. In addition, we carefully selected phospholipids to, not strictly match membrane compositions which is very difficult regarding their variable compositions in time and space, but at least to work with realistic lipid bilayers for eukaryotic (BLT2) or prokaryotic (OmpX) MPs.

2) While the authors promote the presence of just the three ^{13}C NMR probes in BLT2 as an asset to their study (BLT2 has only 2 Met and 1 Ile), I can't help thinking that introducing this promising technology would have made a more convincing case if conducted on a well-characterised MP such as e.g. OmpX or similar. The complete assignments available in the literature would have allowed a much more global and subtle mapping of the effects investigated. As such the paper remains a mixture of introducing a promising technology while illustrating it through the use of a less than convincing target where only a low resolution view of this multispan protein is offered. Under the given circumstances we don't know for example what effect pressure and lipid choice exert e.g. on intra/extra membrane regions including the orthosteric binding pocket. The authors emphasize the ability to discriminate between multiple conformations for each of the three probes as a result of using highly deuterated BLT2 receptor, but this is beyond the point and the real issue is that only very little of the protein is covered while there remain vast 'dark regions' over which we don't know how they respond to pressure. As the authors frequently refer to the important phenomenon of void volume redistribution this is important. I do fully understand that the authors wanted to increase the impact and timeliness of the manuscript by using a GPCR but I think that the latter circumstances due to the sparse coverage not only take away from this otherwise carefully conducted investigation but also introduce an element of ambiguity.

Answer: we would like to thank the reviewer for this comment and suggestion. In the meantime, we have performed new experiments with OmpX which undeniably reinforces the generality, quality and relevancy of our study. To do so, we performed NMR experiments with ^{15}N and methyl labeled and deuterated OmpX in DMPC nanodiscs. The choice of this lipid is based on two considerations:

1) the acyl chains of the lipids of the outer membrane of *E. coli* are predominantly made with 14 carbon atoms. This is one of the reason why Omp proteins have relatively short hydrophobic TM region (about 20–24 Å) compared to the TM regions of α -helix-bundle membrane proteins (~30–32 Å) found in the inner membrane (e.g. see Kleinschmidt & Tamm, *J. Mol. Biol.* (2002) 324, 319–330).

2) DMPC is the lipid that has been used in the literature of NMR studies of OmpX in nanodiscs and the assignment of OmpX in DMPC nanodiscs is available (BMRB code 18796, **ref. 24**).

The analysis of the impact of the dynamics of the lipid bilayer on the protein is based on the following labeling scheme: uniformly deuterated and ^{15}N -labeled and specifically $^{13}\text{CH}_3$ -labeled Ala, Ile ($\Delta 1$) and Val (ProS) residues (following the study from **ref. 35**). As suggested by the reviewer, this protein and labeling scheme allowed us to cover much more spin sites than the GPCR, allowing (for the first time) to probe pressure effects for a large number of positions for a protein immersed in a bilayer.

We also want to emphasize that we deliberately used protonated DMPC instead of the commercially available D54-DMPC which carries perdeuterated acyl chains. This was because the deuteration modifies the transition temperature (see for instance the supplementary figure S7 in **ref. 52**) and pressure. The deuteration of the acyl chains would greatly ease the observation of $^{13}\text{CH}_3$ groups in the protein, but nevertheless we obtained high quality data and were able to follow the evolution of most of the labeled methyl groups we incorporated in OmpX. The study of OmpX and DMPC nanodiscs led to new data that is displayed in **Fig. 1 to 5** associated with new § **page 10** (lines **229-241**) and **pages 11 to 15**.

3) Naturally it is difficult to assess the impact related to the experimental conditions where hydrostatic pressure acts omnidirectional, while a physiological membrane is predominantly exposed to lateral pressure effects and changes across the bilayer. We don't know how this fundamental difference in how pressure is applied manifests itself on lipids and protein, and the authors should add this as a caveat in their discussion.

Answer: the reviewer underlies an important point.

Hydrostatic pressure is a thermodynamical parameter and the membrane lateral pressure is a physical parameter of the membrane defined as the depth-dependent distribution of lateral stresses within the bilayer. Like the other fundamental orthogonal thermodynamical variable, temperature, the hydrostatic pressure can modify the membrane lateral pressure or tension by acting on the different contributions that define this membrane property, notably the membrane thickness. These contributions are: the electrostatic repulsion and the level of hydration of lipid heads, and the van der Waals interactions. If we refer to the literature, in the reasonable pressure range used in our study, just the latter are noticeably impacted. Therefore, to our opinion, hydrostatic pressure and lateral pressure are essentially unrelated, although, of course, the isotropic and macroscopic hydrostatic pressure applied to highly hydrated membrane very likely impacts the microscopic and local lateral pressure existing in membranes. To clarify this point, we added few sentences on **lines 262-267** (sentence that is also related to point N°5 raised by the reviewer, *vide infra*) and lines **351-354**.

Specific:

4) p4, 71. The authors claim that the presence of only 2 methionine and isoleucine allows the exploration of the receptor conformational landscape with great accuracy (see above). This is misleading as the presence of only 3 active NMR probes provides the receptor landscape at very low resolution. This needs rephrasing to take this into account.

Answer: as requested, the following sentence has been substituted (lines **434-435**):

“Hence, it represents an ideal candidate to explore with a great accuracy the impact of the dynamic of lipids on a membrane protein conformation space.”

by :

“Although NMR derived dynamic information will be restricted locally around these 3 residues, this opens the possibility to explore the impact of the dynamics of lipids on three strategic locations in the trans-membrane region of this protein which exhibits a complex conformational space.” (lines **435 to 438**).

5) p8, 181: The authors state that the effect of the receptor under pressure parallels the effect of cholesterol. While this is the case for DPPC, however, it seems to achieve the opposite effect for the POPC/POPG system. The current statement gives the impression that both bilayers behave in analogy as if cholesterol was added.

Answer: our sentence was confusing and we apologize for that. Indeed, while the interpretation of the effect of the cholesterol was correct in the fluid phase, this was not the case concerning the gel phase. Now this part of the text has been removed (originally on page 8, lines 188-189 : “, and below T_m it transforms the liquid-ordered phase into a gel phase [11].” And we added the following sentence (now on **page 11, lines 262-267**):

“In the gel phase, the addition of the receptor may accentuate some changes in the physical properties of the lipid bilayer that already lead to a gelation of the phase upon pressurization. This may concern a more pronounced increase of the membrane thickness or decrease in the lipid area and volume. For instance, the membrane thickness and hydrostatic pressure can both influence the lipid bilayer lateral tension which may in turn further slow down the dynamics of DPPC from head to toes [36].”

6) p12, 313 paragraph: The authors interpret the changes in volume between substates in a manner that suggests that agonist binding leads to a substantial void volume redistribution. In this context the appearance of additional cavities in the pre-active state needs to be commented on while taking into account more recent indications in the literature that such cavities are likely to be filled with water molecules.

Answer: this is an important remark.

In the case of BLT2, the population of the low-populated pre-active (*PA*) states decreases with pressure. We also observed in our study that the population of *PA* states are negatively correlated to the increase of the gelation of the lipid bilayer upon pressurization. This is why these states are almost not visible at atmospheric pressure in DPPC bilayer compared to the fluid POPC/POPG mix in accordance with nucleotide exchange assay experiments and also why this population decreases with pressure in POPC/POPG. In other words, the increase in the lipid gelation would exert an opposite effect of the *PA* states compared to other studies of GPCRs conducted in detergent solutions (**ref. 56 & 57**). And, as already stated in the first version of our manuscript (lines 378-383, and in the new version on **lines 534-537**), the large volume effect of substituting DPPC by POPC/POPG establishes that lipid dynamics controls, to some extent, the conformational energy landscape of BLT2, so the impact of pressure is not limited on the receptor itself.

The present remark of the reviewer pushed us to have a look on the pressure dependence of NMR chemical shifts. Usually NMR resonances show a linear dependence on pressure. Since a change in chemical shift generally corresponds to a change in molecular structure, the linear pressure dependence of the chemical shift means a linear change in protein structure as a function of pressure. If a nonlinearity is observed for some residues, this means that for them the compressibility varies with pressure. The inspection of the pressure dependence of side-chain $^{13}\text{CH}_3$ - δ 1-I229 chemical shifts in BLT2 reveals a non-linear evolution with increasing pressure for *PA* sub-states following a quadratic evolution. This nonlinearity was not observed for the initial reservoir of inactive and highly populated conformers (see **Fig. 8**). For M105 and M197, we did not observe a nonlinearity for sub-states *A* to *E*, *i.e.* including *PA* sub-states *C* and *D*. This kind of nonlinear shift has been ascribed in the literature for residues close to water-accessible cavities in soluble and globular proteins, suggesting that the nonlinear shift is related to the hydration of water-accessible cavities (**ref. 54**). Indeed, the number of internal water molecules is expected to increase at high pressures as increasing hydrostatic pressure then forces water molecules into the protein interior, gradually filling cavities. Moreover, it has been observed for globular proteins that water-buried cavities can be enlarged under pressure [**ref. 54**]. Indeed, volume changes due to the hydration of atoms inaccessible in the predominant low energy state and that become exposed to water under hydrostatic pressure lead to a positive contribution to the volume comparable in absolute value to the constrictions of the voids or even greater, explaining why in some cases some low-energy conformations can be stabilized (positive ΔV) by pressurization [**ref. 55**].

To fulfill this point raised by the reviewer, we added few sentences in the manuscript, lines **584-599**.

7) p13, 352. Previous pressure-dependent studies were conducted on detergent solubilised membrane proteins. The authors comment briefly at the comparison of pressure dependent studies in lipid vs detergent, however, their comment is based on the comparison of different receptors. In view of the modality in the energy landscape of different receptors such a comparison will only hold when conducted on the same receptor. Please take this into account and reformulate this part of your discussion.

Answer: related to the previous point (N°6), this paragraph has been modified to emphasize the balance between the effect of the pressure on the receptor itself (void volume distribution and hydration pattern) or via the physical properties of the bilayer [**lines 602 to 606**].

8) The dramatic increases in signal intensity observed on the receptor side as shown e.g. in Figure 3 are rather extraordinary and their cause needs further assessment. While this is potentially interesting and could formulate a tool for receptor studies (the authors mention this in the manuscript) just based on BLT2 there is no indication that other receptors would behave similarly.

This is indeed a very interesting point that will need to be further studied with another alpha-helical membrane proteins. Yet we did not observe the same trend with a quite rigid protein like OmpX for instance, indicating this phenomenon is not general under pressure and may depend on the MP and/or lipid composition. As far as we can tell, the overall shape and oligomerization state of BLT2 nanodiscs are not affected by the pressurization (based on DOSY exp. in the SI) and from additional initial preliminary data we do believe that pressure modifies the exchange rates and/or substate population in BLT2 leading to the observed reduced line-broadening for methyls groups. We believe that the detailed mechanism of this increased sensitivity will deserve a complete investigation, notably by the measurement of relaxation parameters (CPMG, R_2 , R_1), which are notoriously difficult to measure on such challenging samples. We hope we will be able to provide an explanation in another manuscript. To better highlight this point, we decided to move a figure showing this increase in the s/n of BLT2 signals from SI to the manuscript (now **Fig. 9**) along with additional sentences (lines **616-626**).

9) Limited agonist solubility lead to a situation where due to a small amount of ligand the receptor is only partly bound to ligand. How are the ligand equilibria affected by pressure and how does the water/lipid partitioning of the hydrophobic ligand change upon application of pressure? If changes in the ligand-receptor affinity result from the application of pressure then this will lead to changes in the observed signal intensities and question the current interpretation of the spectra.

Whether the pressure affects either the ligand solubility or, most of all, the affinity or both is an interesting point. Regarding the affinity, unfortunately we cannot performed ligand binding experiments at various pressures yet. Nevertheless, from a qualitative point of view, at each pressure, the NMR spectra of BLT2 is different in presence

or in absence of ligand, and notably, peak 2 is absent in all spectra when the agonist is present, while it is visible in *apo* form when BLT2 is in a POPC/POPG nanodisc (e.g. **Fig. S12**). In addition, quantitatively, relative populations of the (inactive) peaks for I229, M105 and M197 are very different at low pressure but also at the highest pressures (**Fig. 7**) precluding a complete disruption of the complex. And this is true for both lipid compositions tested. These observations tell us that the effect of the ligand seems constant between 1 and 2500 bar, albeit some various in affinity might exist.

We added a sentence in the Methods section (lines **797-799**) to summarize the points discussed above.

Reviewer #2 (Remarks to the Author):

In their manuscript entitled “Exploration of the dynamic interplay between lipids and membrane proteins by hydrostatic pressure” Pozza et al describe the influence of hydrostatic pressure on the phase transition behavior of empty lipid bilayer nanodiscs and containing a GPCR in complex with different ligands. The paper addresses the important question of how a membrane protein can influence the phase behavior of the surrounding lipid bilayer environment and vice versa. Since unsaturated native-like lipids show phase transition temperatures that are below 0°C, the use of pressure is an elegant way to capture the properties of the lipid bilayer using NMR spectroscopy performed at ambient temperatures.

I have a few comments and questions that should be considered by the authors:

1.) Integrity of nanodiscs after application of high pressure

Here mostly NMR 1H spectra were used as well as DOSY spectra using the lipid signals as a readout. I am wondering whether other methods have been used to confirm the presented conclusion since I am not sure if NMR is the best method to convincingly show the integrity of nanodiscs. Did the authors try size exclusion chromatography or light scattering experiments (DLS)?

Answer: as suggested, we have performed DLS and gel filtration experiments. We performed these experiments at atmospheric pressure before and after a jump at 2500 bar with POPC/POPG and DPPC nanodiscs but also with DMPC nanodiscs (in relation with additional experiments performed with OmpX suggested by reviewer 1). We let the samples at 2500 bar during 25 min before repeating the measurements at 1 bar. All these experiments showed that nanodiscs pertained their size and polydispersity and hydrodynamic properties after this high pressure stress and confirmed NMR data, *i.e.* that nanodiscs do resist to high pressures (now **Fig. 1** in the manuscript). All that data confirm the excellent stability of these supramolecular structures upon pressurization/depressurization.

2.) Phase transition of empty nanodiscs

The authors mention that DPPC nanodiscs are in the gel phase at any pressure at the chosen temperature of the NMR experiments (25°C). In order to validate the presented NMR method, it would be interesting to see whether the phase transition of DPPC can be captured if the NMR experiments are conducted at a temperature where DPPC is in the liquid crystalline phase (> 40°C). Lipid nanodiscs are stable enough for such an experiment.

Answer: as suggested, we pressurized DPPC nanodiscs at 323K (50°C). The experiments at 50°C show a clear transition along the lipid structure (see **Fig. S2**) confirming it is in the fluid phase at 1 bar and turning into a gel phase at high pressure. The P_m is ~500 bar for CH₂ about 100 bar above the P_m measured with liposomes (see diagram of Fig. 2A). A same series of experiments were performed at 285K, *i.e.* where no fluid-to-gel phase transition is supposed to occur, in order to highlight the transition observed at 323K (**Fig. S2**).

By varying the temperature at atmospheric pressure, we observe a T_m slightly below (~39°C for CH₂) (**Fig. S3**) the value commonly measured with liposomal preparations (41°C). We also performed experiments at 1000 bar where no transition is supposed to occur in order to highlight the transition observed at atmospheric pressure (**Fig. S3**). This slightly lower T_m value observed is in accordance with the slightly higher P_m value obtained as lowering the temperature is equivalent, at a first approximation, to increase pressure.

Legend to Figure 1: how were the error bars obtained?

Error bars corresponds to two to three repeated NMR experiments. This is now indicated in the caption of **Fig. 2, line 175**.

The MSP protein that is part of lipid nanodisc particles may also change the phase transition behavior of the lipids. Thus, it would be helpful to compare the presented nanodisc data with liposome preparations. I fully understand the motivation of using nanodiscs for this study since GPCR signals are analyzed here as well. However, for the phase transition behavior studies, a pure lipid system would be an important benchmark.

Answer: we totally agree with this remark. We added in **Fig. S3** the CH₂ and CH₃ temperature phase transitions observed by NMR with liposomes of DPPC. These two curves in blue come from a recent published study of liposomes by NMR spectroscopy (Doyen et al. *Mol Pharm.* **2021** 18(7):2521-2539.) in which the corresponding author also figured in the authorship of the present manuscript (E.L.). In this paper, we showed that NMR could capture at high precision the thermotropic phase transition at 1 bar, which incidentally opened our eyes on the ability of NMR to probe not only pressure induced transition on lipid but also protein behavior.

Again all our data indicate that the lipid phase transition parameters (T_m , P_m) are roughly retained in nanodiscs versus liposomes. Nevertheless, we did not measure at very high precision those parameters, and we cannot exclude some ~ 100 - 200 bar or ~ 1 - 2°C variations.

The new figure seems to show a dramatic difference in cooperativity between liposomes and nanodiscs.

However NMR ¹H signals intensities are highly sensitive to internal lipids dynamics in membrane but also to the overall tumbling time (τ_{auc}) of the particle. The large size difference of the nanodiscs (10 nm) and liposomes (~ 100 nm) used here probably largely impacts ¹H signal intensity making the interpretation of the apparent cooperativity of the transition not straightforward.

3.) Receptor loaded nanodiscs

Since the GPCR (monomer) is relatively large and the authors apparently used MSP1D1 nanodiscs of 10 nm in diameter I am wondering how many lipids are actually present in the loaded nanodiscs and whether the observed effects (in fig. 2) could be caused by a varying degree in lipid content. If considering the surface area of the two lipids used here, it is likely that less POPC is present in the GPCR-containing nanodiscs than DPPC, thus maximizing the effect of the membrane protein.

As mentioned in the Methods section (“Overexpression, Purification and reconstitution in lipid nanodiscs of Human BLT2 receptor”), “based on the mean surface area per lipid in nanodiscs of 52 \AA^2 for DPPC and 69 \AA^2 for both POPC and POPG, a rough estimation of the number of lipid layers around the receptor is comprised between 2 and 3, considering a cross-sectional area of 140 \AA^2 per transmembrane helix and a total bilayer area of 4400 \AA^2 using the lipoprotein MSP1D1 [ref. 60].” Thus the number of lipids per leaflet is not so different between the two lipid compositions. Indeed, an empirical calculation based on the cross-sectional areas above-mentioned leads to ~ 50 and ~ 66 POPC/POPG and DPPC lipids per leaflet in the presence of the receptor. To highlight this point, we added one sentence in the Methods section that includes also DMPC nanodiscs in the absence or presence of OmpX, **lines 750-753**. hence, as mentioned **line 70** and **lines 692-694**, nanodiscs represent an ideal tool to look at the impact of an integral membrane protein onto lipid dynamics by preventing the dilution of the effect of the receptor into too many lipids.

4.) NMR spectra of the GPCR

How can the authors exclude the occurrence of partially unfolded or misfolded species of the GPCR preparation that might also contribute separate signals in the ¹³C isoleucine or methionine NMR spectra and potentially might respond to changes in pressure?

By themselves methyl NMR crosspeaks do not indeed unambiguously report on the folded/unfolded nature of the protein. What we can say is that our preparations are highly reproducible on an NMR point of view at 1 bar but also at high pressure and that former quantitative measurements based on ligand binding experiments indicated that almost 100% (Fig. 1a in **Ref. 52**) of the receptor was fully active and stable in lipid nanodiscs, suggesting that the protein is fully folded at 1 bar. The various crosspeaks communicate either through ligand binding or under pressurization and pressure does not lead to the appearance of new protein crosspeaks in the spectra, which is not in favor of protein unfolding. Pressure is known to unfold hydrosoluble protein but to our

knowledge there is no report yet of protein unfolding in membrane with pressure. Here we believe that protein unfolding would lead to a dramatic spectral change, not only in ^1H - ^{13}C spectra but also in ^1H - ^{15}N spectra which was not visible. Our guess is that protein unfolding in membrane would be essentially irreversible leading to aggregation, but this needs to be confirmed. This is why we believe that the NMR signals all correspond to the well-folded states of the receptor populated at 1 bar, and that pressure simply mildly modifies their populations.

Could the higher intensity of the NMR signals at high pressure also be explained by a better homogeneity or even the presence of smaller nanodiscs?

Indeed, we explored various hypothesis for the higher intensity. At the high nanodisc concentration used here, nanodisc/receptor may self-associate and pressure could have disrupted the large assembly. Nevertheless the DOSY experiments tell us that the hydrodynamic properties (translational diffusion) and therefore the apparent radius of the nanodiscs is not particularly affected by pressurization. At the present stage of our investigations, we prefer to invoke a variation in the $^{13}\text{CH}_3$ chemical exchange regime or an improved homogeneity of the receptor in the nanodisc as proposed by the reviewer that become more favorable at high pressures (see also our answer to the point N°8 raised by reviewer 1).

Fig 3B: Why does the lower-field signal at ~1 ppm the 1D spectrum in red show a dispersive line shape or is even negative (blue spectrum on the right)?

Thank you for his remark. This dispersive signal corresponds to t_1 noise arising from the intense ^1H signal of lipid methylene groups, and is therefore a spectral artefact, with no impact of spectral quality and on interpretation of signals remote from the artefacts. This information has now been indicated in the legend of the figure (now **Fig. 6** in the revised version).

5.) The authors should explain in more detail what novel insights are presented in the manuscript since many aspects of lipid phase transition behavior and the effect of incorporated membrane proteins have been explored already.

The novel insights provided here essentially reside in the very high resolution at either the lipid or MP scale. To our knowledge, there is no study in the literature that reports, by NMR or other method, on such degree of detail in the dynamic interplay between lipids and MPs. And this is thanks to the association of 3 components: 1) pressure; 2) nanodiscs; 3) solution-state NMR. The association of the three above-mentioned points allows to obtain high resolution spectra below and above the main gel-to-fluid main phase transition. This gives rise to a fine characterization of the lipid dynamics and in the presence or not of an integral membrane protein at an atomic level with a great accuracy. Indeed, the effect of the phase lipid transition can be followed for each proton from phospholipids. Moreover, the effects of the lipid phase transition on the proton chemical shift depend on the proton position along the phospholipid chain, and report on the bilayer depth. In addition, the effects of inserted protein into the lipid bilayer depend on the fatty acid chain length. Although this latter aspect has already been observed, in our study we give an atomistic view by following the signal variation of each proton position along the lipid tails and heads.

1) pressure: first, as mentioned by reviewer 1, this "is an elegant way to capture the properties of the lipid bilayer using NMR spectroscopy performed at ambient temperatures". Indeed, in the context of lipid bilayers, pressure has the great advantage to be compatible with high-resolution solution-state NMR spectroscopy to studying phase transition with T_m below 0°C . Second, pressure as a thermodynamic variable has the advantage to induce volume variations only by contrast with temperature that induces both volume and thermal energy which complicates the thermodynamic analysis.

2) nanodiscs: these nanometric lipid bilayers ensure the acquisition of highly resolved NMR signals and prevent the dilution of the effect of the MP into too many lipids.

3) solution-state NMR compared to solid-state NMR. Usually, solid-state studies are based on a pair of isotopes from separate experiments which have different properties including sensitivity and relaxation. Here, the behavior of lipid phase transition and the lipid-protein interplay are probed simultaneously by detecting the same type of nuclei (^1H and ^{13}C) on the same sample. But above all, solution-state NMR associated with nanodiscs offer a resolution never achieved in the field.

To summarize all these points, we added in the conclusion few sentences (**page 26**, lines **688-690 & 695-699**).

6.) Minor point: The large number of supporting information figures renders this manuscript quite hard to read. Thus, the authors might consider including some of the data into the main figures or combine data.

Despite the additional experiments conducted with OmpX or OmpX-free DMPC nanodiscs, we succeeded to reduce the number of figures in SI (initially 22 figures, now 15) by either integrating some of them in the main manuscript or removing some.

Additional modifications:

1. We noticed one mistake in the original Fig.~2: the panel describing protons 15,17 (down the left corner) was identical to the panel of protons 14,16. This is due to an error when we built the figure. Now in the new Fig.~2 this has been corrected. Fortunately, this has no incidence on the interpretation of the data as the evolution of protons 15,17 and 14,16 are quite similar in the absence or presence of the receptor.

2. One legend concerning POPC/POPG HOLO was missing at the bottom of Fig.~4 (now Fig.~7). This has been corrected.

REVIEWERS' COMMENTS

Reviewer #1 (Remarks to the Author):

The authors have provided a revised manuscript where by and large they have addressed my queries. A lot of new information from a second protein was generated and included in the revised manuscript. By including substantial amounts of additional new data they have not only responded to the reviewer's criticism but have successfully extended the basis of their findings and have widened the scope of their original study. I can now recommend this manuscript for publication.

Reviewer #2 (Remarks to the Author):

The authors did a fantastic job in addressing all of my previous concerns and suggestions. Therefore, publication of the manuscript is now strongly recommended.

The only thing that caught my eye is the relatively low spectral quality of the ^{15}N -SoFast-HMQC experiments of OmpX in lipid nanodiscs with a large number of overlapping peaks in the random coil region. We usually observe such spectral features with OmpX that was refolded with only washed inclusion bodies without further purification. Since overlapping peaks have not been used by the authors for their analysis, no further experiments are requested. However, the spectral features need to be described and the purity of the OmpX nanodisc preparation should be visualized by SDS-PAGE.

POINT-BY-POINT ANSWER TO REVIEWER COMMENTS

We would like to thank both reviewers for having positively considered our work and allowing its future publication in Nature Communications. We are also grateful to the editor for having given us enough time to complete this work.

REVIEWER COMMENTS

Reviewer #1 (Remarks to the Author):

The authors have provided a revised manuscript where by and large they have addressed my queries. A lot of new information from a second protein was generated and included in the revised manuscript. By including substantial amounts of additional new data they have not only responded to the reviewer's criticism but have successfully extended the basis of their findings and have widened the scope of their original study. I can now recommend this manuscript for publication.

Thank you for these positive comments and your recommendation for publication in Nat Commun.

Reviewer #2 (Remarks to the Author):

The authors did a fantastic job in addressing all of my previous concerns and suggestions. Therefore, publication of the manuscript is now strongly recommended.

Thank you! Despite the difficult global health situation, which sometimes delayed our work, we enjoyed studying OmpX under pressure. It was very instructive for us and opened up many perspectives.

The only thing that caught my eye is the relatively low spectral quality of the ^{15}N -SoFast-HMQC experiments of OmpX in lipid nanodiscs with a large number of overlapping peaks in the random coil region. We usually observe such spectral features with OmpX that was refolded with only washed inclusion bodies without further purification. Since overlapping peaks have not been used by the authors for their analysis, no further experiments are requested. However, the spectral features need to be described and the purity of the OmpX nanodisc preparation should be visualized by SDS-PAGE.

The poor resolution for cross-peaks in the center of the 2D ^1H , ^{15}N spectra displayed in Fig. 3c is likely due to the fast $^1\text{H}^{\text{N}}$ -solvent exchange for the flexible residues in OmpX loops at pH 7.3 and 25°C and 40°C. The solvent/ $^1\text{H}^{\text{N}}$ exchange rate of flexible and solvent accessible residues is usually rather rapid at pH > 7 and $T \geq 25^\circ\text{C}$ and under these conditions severe line broadening usually occurs, notably in the ^{15}N dimension of non-refocused ^1H - ^{15}N experiments.

By the way, as requested by reviewer 2, we added in Fig. S6 two polyacrylamide SDS-PAGE gels, one showing the purity of inclusion bodies and the second the quality of our NMR sample. The key feature in obtaining high-quality IBs lies in the selection of the right colony on plate, i.e. among the ones that over-express the most the outer membrane protein. A very high yield of expression guarantee a high degree of purity of subsequent IBs, as attested by one of the gels displayed in Fig. S6.